# Enhancing CRISPR-Cas9 gRNA efficiency prediction by data integration and deep learning

Xi Xiang[1,2,3,4,12], Giulia I. Corsi [5,12], Christian Anthon[5,12], Kunli Qu[1,6,12], Xiaoguang Pan[1], Xue Liang[1,6], Peng Han[1,6], Zhanying Dong[1], Lijun Liu[1], Jiayan Zhong[7], Tao Ma[7], Jinbao Wang[7], Xiuqing Zhang[3], Hui Jiang[7], Fengping Xu[1,3], Xin Liu [3], Xun Xu [3,8], Jian Wang[3], Huanming Yang[3,9], Lars Bolund[1,3,4], George M. Church[10], Lin Lin [1,4,11], Jan Gorodkin[5,13✉] & Yonglun Luo [1,3,4,11,13✉]

The design of CRISPR gRNAs requires accurate on-target efficiency predictions, which demand high-quality gRNA activity data and efficient modeling. To advance, we here report on the generation of on-target gRNA activity data for 10,592 SpCas9 gRNAs. Integrating these with complementary published data, we train a deep learning model, CRISPRon, on 23,902 gRNAs. Compared to existing tools, CRISPRon exhibits significantly higher prediction performances on four test datasets not overlapping with training data used for the development of these tools. Furthermore, we present an interactive gRNA design webserver based on the CRISPRon standalone software, both available via https://rth.dk/resources/crispr/. CRISPRon advances CRISPR applications by providing more accurate gRNA efficiency predictions than the existing tools.

[1] Lars Bolund Institute of Regenerative Medicine, Qingdao-Europe Advanced Institute for Life Sciences, BGI-Qingdao, Qingdao, China. [2] BGI Education Center, University of Chinese Academy of Sciences, Shenzhen, China. [3] BGI-Shenzhen, Shenzhen, China. [4] Department of Biomedicine, Aarhus University, Aarhus, Denmark. [5] Center for non-coding RNA in Technology and Health, Department of Veterinary and Animal Sciences, Faculty of Health and Medical Sciences, University of Copenhagen, Frederiksberg, Denmark. [6] Department of Biology, University of Copenhagen, Copenhagen, Denmark. [7] MGI, BGI-Shenzhen, Shenzhen, China. [8] Guangdong Provincial Key Laboratory of Genome Read and Write, BGI-Shenzhen, Shenzhen, China. [9] Guangdong Provincial Academician Workstation of BGI Synthetic Genomics, BGI-Shenzhen, Shenzhen, China. [10] Department of Genetics, Blavatnik Institute, Harvard Medical School, Boston, MA, USA. [11] Steno Diabetes Center Aarhus, Aarhus University, Aarhus, Denmark. [12] These authors contributed equally: Xi Xiang, Giulia I. Corsi, Christian Anthon, Kunli Qu. [13] These authors jointly supervised this work: Jan Gorodkin, Yonglun Luo. ✉email: gorodkin@rth.dk; alun@biomed.au.dk

Clustered Regularly Interspaced Short Palindromic Repeats (CRISPR)-associated protein 9 (Cas9) has been successfully harnessed for programmable RNA-guided genome editing in prokaryotes, humans and many other living organisms[1–5]. A successful CRISPR gene editing application depends greatly on the selection of highly efficient gRNAs. Several machine and deep learning methods have been developed in the past decade to predict on-target gRNA activity[6–16]. However, some of these models exhibit discrepancies in the parameters selected for model validation, and in the data used for testing, which directly impact on the performances reported for such tools (Supplementary Notes 1-2). For instance, the prediction performances of the recent DeepSpCas9variants model[7] appear to be substantially higher when both canonical and noncanonical PAMs are employed for testing compared to an evaluation based solely on canonical PAMs, which are preferred for gRNA designs (Spearman's R = 0.94 decreases to R = 0.70, Supplementary Fig. 1). While the application of more advanced machine learning strategies has relatively modest impact on gRNA activity prediction performances, a significant improvement can be achieved by increasing the size and the quality of the training data (Supplementary Note 1).

Recent models trained on large-scale data still lack full saturation of their learning curve[9,14], thus leaving space for further data-driven improvement. At present, the amount of gRNA efficiency data suitable to develop machine learning models remains scarce, mostly due to the low homogeneity between studies in terms of experimental design and cleavage evaluation methodologies, which can vary from loss of function, e.g., Xu et al. (2015), Hart et al. (2015), and Doench et al. (2014–2016)[14,17–19], to indels quantification, e.g., Chari et al. (2015), Wang et al. (2019), and Kim et al. (2019–2020)[7–9,20,21]. It is thus essential to produce additional data from gRNA activity compatible with previous studies to develop more accurate prediction methods. To overcome the scarcity of experimental on-target efficiency data previous studies have employed techniques such as data augmentation, widely known in the field of image recognition, creating new input–output pairs by introducing minor alterations in the input sequence of experimentally validated gRNAs while considering their output, the efficiency, unaffected[11]. However, while two mirrored images are encoded by highly different input matrices but maintain the same original meaning, augmented gRNA data are highly redundant and do not guarantee consistency in terms of cleavage efficiency. Thus, data quantity remains the major bottleneck for improving predictors[9,14] (see also Supplementary Note 1).

Here, we show that lentiviral surrogate vectors can faithfully capture gRNA efficiencies at endogenous genomic loci. Using this approach, we generate on-target gRNA activity data for 10,592 SpCas9 gRNAs. After integrating them with complementary published data (resulting in activity data for a total of 23,902 gRNAs), we develop a deep learning prediction model, CRISPRon, which exhibits significantly higher prediction performances on independent test datasets compared to existing tools. The analysis of features governing gRNA efficiency shows that the gRNA-DNA binding energy $\Delta G_B$ is a major contributor in predicting the on-target activity of gRNAs. Furthermore, we develop an interactive gRNA design webserver based on the CRISPRon standalone software, both available via https://rth.dk/resources/crispr/. The software may also be downloaded from GitHub on https://github.com/RTH-tools/crispron/[22].

## Results and discussion
### Massively parallel quantification of gRNA efficiency in cells.
To generate further high-quality CRISPR on-target gRNA activity data, we established a high-throughput approach to measure gRNA activity in cells (Fig. 1a) based on a barcoded gRNA oligonucleotide pool strategy as described previously[23,24]. Several optimizations of the original methods[23,24] were introduced to simplify and streamline vector cloning, lentiviral packaging and enrichment of gene edited cells (see Supplementary Note 3, Supplementary Fig. 2). To validate if the indel frequency introduced at the 37 bp surrogate target site could recapitulate that at the corresponding endogenous sites, we analyzed indel frequency at 16 surrogate sites and their corresponding endogenous genomic loci in HEK293T cells by deep sequencing. We obtained a fine correlation between the surrogate and endogenous sites in terms of indel frequencies and profiles (Supplementary Fig. 3, Spearman's R = 0.72, p-value = 0.0016), in agreement with previous findings[8,9,23,24].

We next generated a large dataset of high-quality CRISPR gRNA activity data in cells using this optimized approach. A pool of 12,000 gRNA oligos, targeting 3834 human protein-coding genes (Supplementary Data 1, Supplementary Note 4), were array-synthesized and selected to avoid large overlap with existing datasets. Targeted amplicon sequencing (depth > 1000) of the surrogate oligo pool, surrogate gRNA plasmid library and transduced wild-type HEK293T cells (multiplexity of infection (MOI) of 0.3) revealed that over 99% of the designed gRNAs were present in the 12 K gRNA plasmid pool and transduced cells (Supplementary Figs. 4-5, source data). We transduced the SpCas9-expressing and wild-type HEK293T cells with the gRNA library with a MOI of 0.3 and a transduction coverage of ~4000 cells per gRNA. A pipeline was established to analyze CRISPR-induced indels and remove sequence variants introduced by oligo-synthesis, PCRs, and deep sequencing, as well as low quantity sites (less than 200 reads, see Methods). Indel frequencies in the cells 2, 8, and 10 days after transduction were analyzed by targeted deep sequencing (Supplementary Fig. 6). Following increased editing time and enrichment of edited cells (puromycin selection), indel frequency rose significantly in cells from day 2 to day 8–10 (Fig. 1b). Overexpression of SpCas9 by doxycycline (Dox) addition leads to a skewed distribution of gRNA efficiency (Supplementary Fig. 7, Supplementary Note 4), thus gRNA efficiencies from Dox-treated SpCas9 cells were excluded for gRNA efficiency prediction model establishment. The indel frequency (on-target activity) of gRNAs from day 8 and 10 were well correlated (Fig. 1c, Spearman's R = 0.91). Corroborating previous findings, the indel types introduced by SpCas9 comprise mainly small deletions and 1 bp insertion (Fig. 1d, Supplementary Figs. 7-8) and compared to day 2 the indel types from day 8–10 are better correlated with the indel profiles predicted by inDelphi[24] (Fig. 1e, Supplementary Fig. 7-8, Supplementary Note 5), a machine learning algorithm for predicting CRISPR-induced indels. Our data also revealed that the inserted nucleotide of the most frequent indel type (1 bp insertion) is most frequently the same as N17 nucleotide of the protospacer (4 bp upstream of the PAM) (Fig. 1f, Supplementary Fig. 7, Supplementary Note 5). The average gRNA activity from day 8 and 10 was used for subsequent analyses and model establishment. As a result, we obtained high-quality gRNA activity data for 10,592 gRNAs, of which 10,313 gRNAs are unique for this study (Supplementary Fig. 9, Supplementary Data 1). To independently validate the CRISPR gRNA activity captured by the lentiviral surrogate vector library, we compared gRNA efficiencies commonly measured in our study to those of Kim et al. (2019) and Wang et al. (2019)[8,9] (Fig. 1g). We observed a good correlation (Spearman's R = 0.67 to both) between gRNA activities measured by our study and others, higher compared to the agreement between these two existing protocols (Spearman's R = 0.52). Our gRNA efficiency data match characteristics of

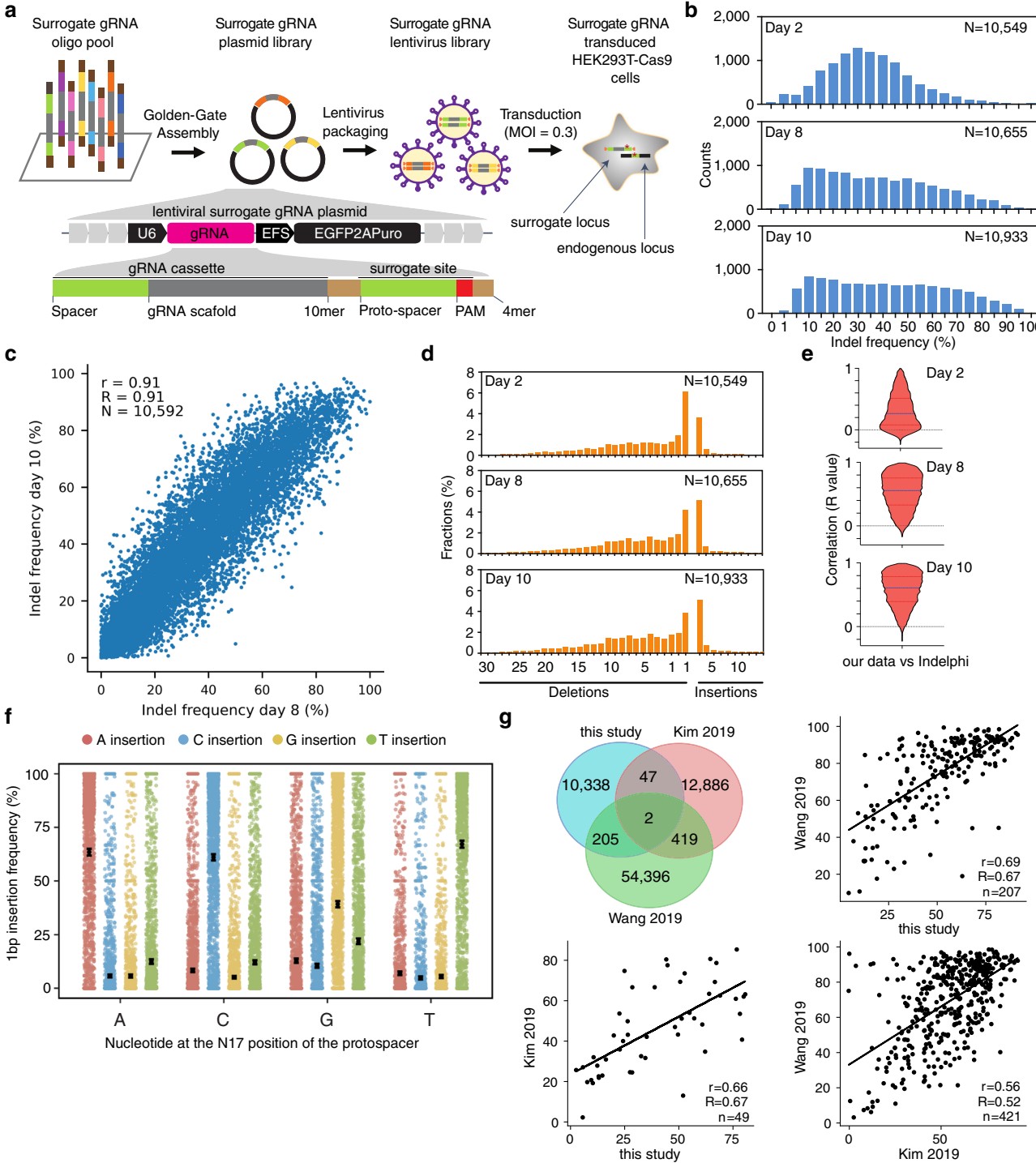

**Fig. 1 High-throughput quantification of gRNA efficiency in cells. a** Schematic illustration of the lentiviral surrogate vector, oligo pool synthesis, PCR amplification, golden-gate assembly, lentivirus packaging, and transduction. **b** gRNA editing efficiency of all surrogate sites measured by targeted amplicon sequencing. Results are shown for HEK293T-SpCas9 cells at 2, 8, and 10 days after transduction. **c** Correlation between gRNA editing efficiency at 8 and 10 days after transduction. **d** Indel profiles (1–30 bp deletion, 1–10 bp insertion) for all surrogate sites introduced by SpCas9 in HEK293T-SpCas9 cells at 2, 8, and 10 days post transduction. **e** Correlation between the indel profiles measured in cells and those predicted by inDelphi. Data are presented as violin plot with median and quartiles. **f** Dot plot of 1-bp insertion indel frequency (mean ± 95% confidence interval), stratified by the nucleotide at N17 position of the protospacer and the type of nucleotide inserted (see also Supplementary Fig. 7). **g** Correlation between gRNA editing efficiencies measured in this and other major studies for common gRNA + PAM (23 nt) examples, also displayed in a Venn diagram.

previous findings, with a preferential range of GC content between 40 and 90%[25] and stable gRNA structures being unfavorable, in particular for minimum folding energies (MFE) < −7.5 kcal/mol[26] (Supplementary Fig. 10). We conclude that the high-quality gRNA activity dataset of 10,592 gRNAs measured in

cells by our study represents a valuable source to further improve the quality of CRISPR-gRNA designs.

**Enhanced gRNA efficiency prediction.** We developed a deep learning model, which combines sequence and thermodynamic

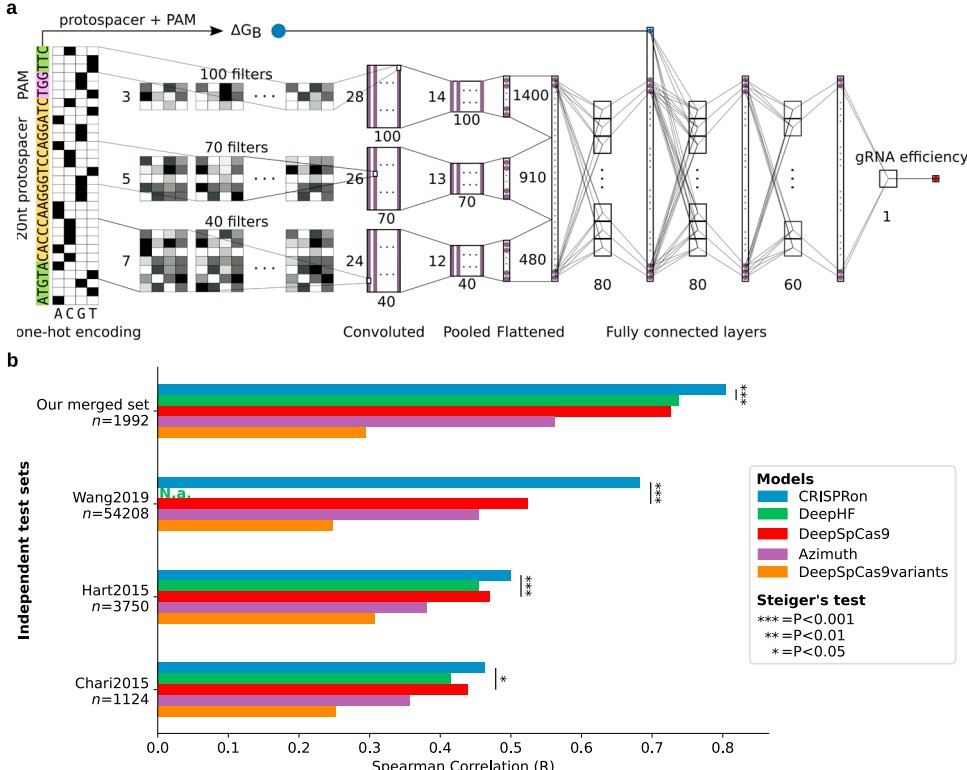

**Fig. 2 The CRISPRon model and generalization ability on independent test sets. a** Schematic representation of the CRISPRon input DNA sequence and prediction algorithm. The inputs to the deep learning network are the one-hot encoded 30mer and the binding energy ($\Delta G_B$). Note that only the filtering (convolutional) layers and the 3 fully connected layers are shown explicitly and that the thin vertical bars are the output of one layer, which serves as input for the next layer. **b** Performance comparison between CRISPRon and other existing models on independent test sets larger than 1000 gRNAs. N.a. not available (all gRNAs were regarded as training data due to lack of explicit train-test separation). CRISPRon_v0 was employed for testing on the internal independent test set ("Our merged set", including gRNAs from both our study and Kim et al. (2019)). CRISPRon_v1, or simply CRISPRon, was used for the external independent test sets (for a description of the CRISPRon versions, see Supplementary Table 1). The two-sided Steiger's test P-values of all comparisons are reported in Supplementary Data 2.

properties automatically extracted out of a 30 nt DNA input sequence constituted of the protospacer, the PAM and neighboring sequences for precise gRNAs activity predictions (Fig. 2a). In addition to the sequence composition, the model embeds the gRNA-target-DNA binding energy $\Delta G_B$, described by the energy model used in CRISPRoff[27], which encapsulates the gRNA-DNA hybridization free energy, and the DNA-DNA opening and RNA unfolding free energy penalties. $\Delta G_B$ was observed to be a key feature for predicting on-target gRNA efficiency (see Supplementary Note 6 and feature analysis below). We first trained deep learning models solely on our dataset (Supplementary Table 1) and compared their predictions with those of existing tools on both internal and external independent test datasets. To do that, our CRISPR gRNA activity data were carefully partitioned into six subsets ensuring clustering of the closest gRNA sequences within the same partition (see Methods). The first model, pre-CRISPRon_v0, was trained with a 5-fold cross-validation while using a 6th partition as an internal independent test set solely for measuring the performance. The pre-CRISPRon_v0 and DeepSpCas9 models displayed remarkable and comparable generalization ability when tested on data from the study of one another (Spearman's R > 0.70 for both), confirming our data and Kim et al. (2019) data as highly compatible (Supplementary Data 2). The second model (pre-CRISPRon_v1, see Supplementary Table 1 and Supplementary Data 2) was constructed to evaluate on external independent test sets by training on all six partitions with a 6-fold cross-validation. This model displayed performances similar to those of existing tools.

Since pre-CRISPRon_v0 and DeepSpCas9 held comparable performances when trained on their respective datasets, we fused our data with that of Kim et al. (2019) using a linear rescaling based on the 30mer sequences found in both datasets, resulting in a dataset of 23,902 gRNAs (30mer, Supplementary Fig. 9). We did not fuse with the datasets measuring efficiency as indel frequency of Wang et al. (2019) and Kim et al. (2020), because of their scarce coverage of the general gRNA activity landscape (Supplementary Note 2 and 7). After dividing the joint dataset of our study and Kim et al. (2019) into six partitions as explained above, we first developed CRISPRon_v0 with a 5-fold cross-validation to evaluate the model on the internal independent test set. The CRISPRon_v0 increased the performance over pre-CRISPRon_v0 on the Kim et al. (2019) dataset, while only a minor loss (<0.025 in Spearman's R) was observed on our data (Supplementary Table 1 and Supplementary Data 2). On the internal independent test set, CRISPRon_v0 exceeded the performance (Spearman's R = 0.80) of notable predictors, such as Azimuth (R = 0.56), DeepSpCas9 (R = 0.73), DeepHF (R = 0.74), and DeepSpCas9variants (Fig. 2b, Supplementary Fig. 11). A final model, CRISPRon_v1 (hereafter called CRISPRon, see Supplementary Table 1 and Supplementary Data 2), was then trained on the full combined dataset with a 6-fold cross-validation. External independent test sets with more than 1000 gRNAs (Fig. 2b) were employed for testing while again ensuring no overlap between what the respective models were trained on (see Methods). On these external independent test sets CRISPRon achieved the highest prediction performance (R ≈ [0.46, 0.68])

compared to Azimuth (R ≈ [0.36, 0.45]), DeepSpCas9variants (R ≈ [0.25, 0.31]), and to the so far top-performing models DeepSpCas9 (R ≈ [0.44, 0.52]) and DeepHF (R ≈ [0.42, 0.46]). Additional performance evaluations on datasets with less than 1000 gRNAs confirmed CRISPRon as top-performing model (Supplementary Fig. 11, Supplementary Data 2). A web interface for gRNA on-target efficiency predictions with the CRISPRon model is made available via https://rth.dk/resources/crispr/. The webserver interface utilizes the IGV javascript plugin available from github[28].

**Features important for predicting gRNA efficiency**. To characterize the gRNA features with the highest impact on gRNA efficiency predictions we trained a gradient boosting regression tree (GBRT) model based on the combined data from our study and Kim et al. (2019) and applied two methods for feature analysis: the Shapley Additive exPlanations (SHAP)[29] and the Gini importance[30] (Supplementary Fig. 12, full details in Supplementary Note 6). Both methods highlight that thermodynamic properties, above all $\Delta G_B$, give a considerable contribution to the learning process. The most notable sequence-composition features include the two nucleotides proximal to the PAM, where G and A are favored over C and T and the presence of the dinucleotide TT, which relates with weak binding free energies and is unfavorable.

**Limitations to the study**. A few limitations of using the lentiviral surrogate vectors to capture CRISPR gRNA efficiency are highlighted for the need of future improvements. The DSBs generated by CRISPR-Cas9 are predominantly repaired by the non-homologous end joining (NHEJ) and microhomology-mediated end joining (MMEJ) pathways, which leads to the introduction of small indels at the DSB site. However, large deletions or chromosomal rearrangements have also been reported in CRISPR editing as outcomes of repaired mediated by e.g., homology-directed repair (HDR) or single-strand annealing (SSA) in cells[31,32]. The gRNA efficiency quantification approach in this study is based on a 37 bp surrogate target site. Thus, SpCas9 editing outcomes such as large deletions or chromosomal rearrangements are not captured by our method. Earlier, we have discovered that chromatin accessibility at the editing sites affects CRISPR gene editing efficiency[26]. Since the 12 K lentivirus library was randomly inserted in the genome of the targeted cells, the chromatin accessibility state of the surrogate site might be different from the endogenous target site.

**Concluding remark**. In summary, we report on the generation of on-target gRNA activity data for 10,592 SpCas9 gRNAs and the development of a deep learning model, CRISPRon, which exhibits more accurate gRNA efficiency predictions than other existing tools.

## Methods

**DNA vectors**. The 3rd generation lentiviral vector backbone was generated by synthesis (Gene Universal Inc) and cloning. The human codon-optimized SpCas9 expression vector was based on a PiggyBac transposon vector, carrying a hygromycin selection cassette. All DNA vectors have been Sanger sequenced and can be acquired from the corresponding author YL's lab. The lentiviral vector generated by this study for cloning surrogate oligos has been made available through Addgene (plasmid # 170459). A detail protocol is also made available at the shared protocols platform[33].

**Design of the 12 K surrogate oligo pool**. Each oligo consists of the BsmBI recognition site "cgtctc" with 4 bp specific nucleotides "acca" upstream, following the GGA cloning linker "aCACC", one bp "g" for initiating transcription from U6 promoter, 20 bp gRNA sequences of "gN20", 82 bp gRNA scaffold sequence, 37 bp surrogate target sequences (10 bp upstream sequences, 20 bp protospacer and 3 bp

PAM sequences, 4 bp downstream sequence), the downstream linker "GTTTg", and another BsmBI-binding site and its downstream flanking sequences "acgg".

For the 12 K oligo pool was designed as below: (1) Select ~7000 genes from the drugable gene database (http://dgidb.org)[34]; (2) Discard all the exons which the DNA length is less than 23 bp with filtering; (3) Select the first three coding exons of each gene. If the exons number is less than 3, retain all the exons; (4) Extract all the possible gRNA sequences (including the PAM sequence "NGG") in the filtered exons sequence; (5) Look up off-target sites of each gRNA with FlashFry (v 1.80)[35] and discard gRNAs with potential off-target of 0–3 bp mismatches in human genome hg19 and rank each gRNA based on the number of off-target site in an ascending order; (6) Map and extract the 10 bp upstream and 4 bp downstream flanking sequence of each selected gRNA, construct the surrogate target sequence as 10 bp upstream + 23 bp gRNA (include PAM) + 4 bp downstream = 37 bp; (7) Filter out surrogate sites with BsmBI recognition site, because of GGA cloning; (8) Compare all the selected gRNAs with the database of CRISPR-iSTOP[36]; (9) Construct the full-length sequence of each synthetic oligo, which is 170 bp; In total, the 12 K oligos target 3832 genes. The 12 K oligo pools were synthesized in Genscript® (Nanjing, China), and sequences are given in Supplementary Data 1.

**12 K surrogate plasmid library preparation**. First, the 12 K oligos were cleaved and harvested from the microarray and diluted to 1 ng/μl. Next, we performed surrogate PCR1 (Supplementary Data 1). The PCR reaction was carried out using PrimeSTAR HS DNA Polymerase (Takara, Japan) following the manufacturer's instruction. Briefly, each PCR reaction contained 1 μl oligo template, 0.2 μl PrimeSTAR polymerase, 1.6 μl dNTP mixture, 4 μl PrimeSTAR buffer, 1 μl forward primer (10 uM), and 1 μl reverse primer (10 uM) and ddH2O to a final volume of 20 μl.

The thermocycle program was 98 °C 2 min, (98 °C/10 s, 55 °C/10 s, 72 °C/30 s) with 21 cycles, then 72 °C for 7 min and 4 °C hold. To avoid amplification bias of oligos introduced by PCR, we conducted gradient thermocycles and performed PCR products gray-intensity analysis to determine the optimal PCR cycles of 21. The best thermocycles should be in the middle of an amplification curve. In this study, the PCR cycle as 21 for oligos amplification. Instead, for PCR amplification of surrogate sites from cells integrated with lentivirus, the PCR cycle was 25. The final PCR product length was 184 bp. We performed 72 parallel PCR reactions for 12 K oligos amplification, then these PCR products were pooled, and gel purified by 2% agarose gel. One microgram purified PCR product were quantified with PCR-free next generation sequencing (MGI Tech).

The PCR products of 12 K oligos were then used for Golden Gate Assembly (GGA) to generate the 12 K plasmids library. For each GGA reaction, the reaction mixture contained 100 ng lentiviral backbone vector, 10 ng purified 12 K oligos-PCR products, 1 μl T4 ligase (NEB), 2 μl T4 ligase buffer (NEB), 1 μl BsmBI restriction enzyme (ThermoFisher Scientific, FastDigestion) and ddH2O to a final volume of 20 μl. The GGA reactions were performed at 37 °C 5 min and 22 °C 10 min for 10 cycles, then 37 °C 30 min and 75 °C 15 min. Thirty six parallel GGA reactions were performed and the ligation products were pooled into one tube.

Transformation was then carried out using chemically competent DH5a cells. For each reaction, 10 μl GGA ligation product was transformed in to 50 μl competent cells and all the transformed cells were spread on one LB plate (15 cm dish in diameter) with Xgal, IPTG and Amp selection. High ligation efficiency was determined by the presence of very few blue colonies (also see Supplementary Fig. 2). To ensure that there is sufficient coverage of each gRNA of the 12 K library, 42 parallel transformations were performed, and all the bacterial colonies were scraped off and pooled together for plasmids midi-prep. For NGS-based quality quantification of the library coverage, midi-prep plasmids were used as DNA templates for surrogate PCR2, followed by gel purification and NGS sequencing. The PCR primers for surrogate PCR2 are showed in Supplementary Data 1.

**12 K lentivirus packaging**. HEK293T cells were used for lentivirus packaging. All cells were cultured in Dulbecco's modified Eagle's medium (DMEM) (LONZA) supplemented with 10 % fetal bovine serum (FBS) (Gibco), 1% GlutaMAX (Gibco), and penicillin/streptomycin (100 units penicillin and 0.1 mg streptomycin/mL) in a 37 °C incubator with 5% CO2 atmosphere and maximum humidity. Cells were passaged every 2–3 days when the confluence was ~80–90%.

For lentivirus packaging: (Day 1) Wild-type HEK293T cells were seeded to a 10 cm culture dish, 4 × 10^6 cells per dish (10 dishes in total); (Day 2) Transfection. Briefly, we refreshed the medium with 7 mL fresh culture medium to 1 h before transfection (gently, as the HEK293T cells are easy to be detached from the bottom of dish); Next, we performed transfection with the PEI 40000 transfection method. For 10 cm dish transfection, the DNA/PEI mixture contains 13 μg lentiviral 12 K plasmid DNA, 3 μg pRSV-REV, 3.75 μg pMD.2 G, 13 μg pMDGP-Lg/p-RRE, 100 μg PEI 40000 solution (1 μg/μl in sterilized ddH2O), and supplemented by serum-free optiMEM without phenol red (Invitrogen) to a final volume of 1 mL. The transfection mixture was pipetted up and down several times gently, then kept at room temperature (RT) for 20 min, then added into cells in a dropwise manner and mix by swirling gently. (Day 3) Changed to fresh medium; (Day 4) Harvest and filter all the culture medium of the 10 cm dish through a 0.45 μm filter, pool the filtered media into one bottle. Each 10 cm dish generated ~7–8 mL lentivirus crude. Add polybrene solution (Sigma–Aldrich) into the crude virus to a final

concentration of 8 µg/mL. Aliquot the crude virus into 15 mL tubes (5 mL/tube) and store in −80 °C freezer.

**Lentivirus titer quantification by flow cytometry (FCM).** As the 12 K lentiviral vector expresses an EGFP gene, the functional titer of our lentivirus prep was assayed by FCM. Briefly, (1) split and seed HEK293T cells to 24-well plate on day 1, $5 \times 10^4$ cells per well. Generally, 18 wells were used to perform the titter detection, a gradient volume of the crude lentivirus was added into the cells and each volume was tested by replicates. In this experiment, the crude virus gradients were 10, 20, 40, 80, and 160 µl for each well (Supplementary Fig. 5). Another two wells of cells were used for cell counting before transduction; (2) Conduct lentivirus transduction when cells reach up to 60–80% confluence on day 2. Before trans-duction, detach the last two wells of cells using 0.05% EDTA-Trypsin to determine the total number of cells in one well ($N_{initial}$). Then change the remaining wells with fresh culture medium containing 8 µg/mL polybrene, then add the gradient volume of crude virus into each well and swirling gently to mix; (3) On day 3, change to fresh medium without polybrene; (4) On day 4, harvest all the cells and wash them twice in PBS. Fix the cells in 4% formalin solution at RT for 20 min, then spin down the cell pellet at $500 \times g$ for 5 min. Discard the supernatant and re-suspend the cell pellet carefully in 600 µl PBS, and conduct FCM analysis immediately. FCM was performed using a BD LSRFortessa$^{TM}$ cell analyzer with at least 30,000 events collected for each sample in replicates.

The FCM output data was analyzed by the software Flowjo vX.0.7. Percentage of GFP-positive cells was calculated as: Y% = $N_{GFP-positive\ cells}/N_{total\ cells} \times 100\%$. Calculate the GFP percentage of all samples. For accurate titter determination, there should be a linear relationship between the GFP-positive percentages and crude volume. The titter (Transducing Units (TU/mL) calculation according to this formula: TU/mL = ($N_{initial} \times Y\% \times 1000$)/V. V represents the crude volume (µl) used for initial transduction.

**Generation of SpCas9-expressing stable cell lines.** SpCas9-expressing HEK293T (HEK293T-SpCas9) cells were generated by a PiggyBac transposon system. HEK293T cells were transfected with pPB-TRE-spCas9-Hygromycin vector and pCMV-hybase with a 9:1 ratio. Briefly, $1 \times 10^5$ HEK293T cells were seeded in 24-well plate and transfections were conducted 24 h later using lipofectamine 2000 reagent following the manufacturer's instruction. Briefly, 450 ng pPB-TRE-spCas9-Hygromycin vectors and 50 ng pCMV-hybase were mixed in 25 µl optiMEM (tube A), then 1.5 µl lipofectamine 2000 reagent was added in another 25 µl optiMEM and mix gently (tube B). Incubate tube A and B at RT for 5 min, then add solution A into B gently and allow the mixture incubating at RT for 15 min. Add the AB mixture into cells evenly in a dropwise manner. Cells transfected with pUC19 were acted as negative control. Culture medium was changed to selection medium with 50 µg/ml hygromycin 48 h after transfection. Completion of selection took ~5–7 days until the negative cells were all dead in the untransfected cells. The cells were allowed to grow in 50 µg/ml hygromycin growth medium for 3–5 days for further expansion. PCR-based genotyping was carried out to validate the integration of Cas9 expression cassette (Supplementary Data 1). Although the expression of SpCas9 was controlled by a TRE promoter, we observed significant editing efficiency in cells without addition of doxycycline. Thus, the cells were used as a normal SpCas9-expressing model, while SpCas9 overexpression can be induced by Dox induction.

**12 K lentivirus library transduction.** HEK293T-SpCas9 cells were cultured in growth medium with 50 µg/ml hygromycin throughout the whole experiment. For 12 K lentivirus library transduction, (1) on Day −1: $2.5 \times 10^6$ cells per 10 cm dish were seeded (in 12 dishes). For each group, one dish was used for cell number determination before transduction and one dish for drug-resistance (puromycin) test control and the remaining 10 dishes were used for the 12 K lentivirus library transduction (transduction coverage per gRNA exceeds 4000×); (2) Day 0: We first determined the approximate cell number per dish. This was used to determine the volume of crude lentivirus used for transduction using a multiplicity of infection (MOI) of 0.3. The low MOI (0.3) ensures that most infected cells receive only 1 copy of the lentivirus construct with high probability [41]. The calculation formula is: V = N × 0.3/TU. V = volume of crude lentivirus used for infection (ml); N = cell number in the dish before infection; TU = the titter of crude lentivirus (IFU/mL). The infected cells were cultured in a 37 °C incubator; (3) Day 1: 24 h after trans-duction, split the transduced cells of each dish to three dishes equally; (4) Day 2: For the three dishes of split (30 dishes in total, three divided into sub-groups), subgroup 1 (10 dishes) were harvested and labeled as the Day 2 after the 12 K lentivirus library transduction. All cells from this subgroup were pooled into one tube and stored in −20 °C freezer for genomic DNA extraction. The subgroup 2 (10 dishes) was changed to fresh D10 medium contains 50 µg/ml hygromycin + 1 µg/mL puromycin (Dox-free group); The subgroup 3 (10 dishes) was changed to D10 medium contains 50 µg/ml hygromycin + 1 µg/mL puromycin + 1 µg/mL doxycycline (Dox-addition group). (5) The transduced cells were spitted every 2–3 days when cell confluence reaches up to 90%. Cells from Day 2, 8, and 10 were harvested and stored in −20 °C for further genomic DNA extraction. Parallel experiments were performed using wild-type HEK293T cells.

**PCR amplification of surrogate sites from cells.** Genomic DNA was extracted using the phenol-chloroform method. The genomic DNA were digested with RNase A (OMEGA) to remove RNA contamination (In this study, 50 µg RNase A worked well to digest the RNA contamination in 100–200 µg genomic DNA after incubating in 37 °C for 30 min). Then the genomic DNA was purified and sub-jected to surrogate PCR2 (Supplementary Data 1). In this study, 5 ug genomic DNA was used as temperate in one PCR reaction, which contained ~$7.6 \times 10^5$ copies of surrogate construct (assuming $1 \times 10^6$ cells contain 6.6 µg genomic DNA), which covered about 63 times coverage of the 12 K library. In total, 32 parallel PCR reactions were performed to achieve approximately 2016 times coverage of each gRNA and surrogate site. For each PCR reaction, briefly, 50 µl PCR reaction system consists of 5 µg genomic DNA, 0.5 µg PrimeSTAR polymerase, 4 µl dNTP mixture, 10 µl PrimeSTAR buffer, 2.5 µl forward primer (10 uM), and 2.5 µl reverse primer (10 uM) and supplemented with ddH2O to a final volume of 50 µl. The thermo-cycle program was 98 °C 2 min, (98 °C for 10 s, 55 °C for 10 s, 72 °C for 30 s) with 25 cycles, then 72 °C for 7 min and 4 °C hold. Then purify all the PCR products by 2% gel, pool the products together and conduct deep amplicon sequencing.

**Deep amplicon sequencing.** MGISEQ-2000 (DNBseq-G400) was used to perform the amplicons deep sequencing following the standard operation protocol. First, PCR-free library was prepared using MGIeasy FS PCR-free DNA library Prep kit following the manufacturer's instruction. Briefly, measure the concentration of purified PCR products using Qubit 4$^{TM}$ fluorometer (Invitrogen) and dilute the concentration of each sample to 10 ng/µl. Ten microliters diluted PCR product was mixed with an A-Tailing reaction which contained A-Tailing enzyme and buffer, incubated at 37 °C for 30 minutes then 65 °C for 15 min to inactive the enzyme. Then the A-Tailed sample was mixed with PCR Free index adapters (MGI.), T4 DNA Ligase and T4 ligase buffer to add index adapter at both 3' and 5' ends of PCR products. The reaction was incubated at 23 °C for 30 min and then purified with XP beads. Then denature the PCR products to be single-strand DNA (ssDNA) by incubating at 95 °C for 3 min and keep on 4 °C for the subsequent step. Transform the ssDNA to be circles using cyclase (MGI) at 37 °C for 30 min and then digested to remove linear DNA using Exo enzyme at 37 °C for 30 min. Purify the products again by XP beads and assay the concentration of library by Qubit 4 $^{TM}$ fluorometer. The amplicons libraries were subjected to deep sequencing on the MGISEQ-2000 platform. In this study, for each lane four samples (6 ng each) were pooled together for deep sequencing. To avoid sequencing bias induced by base unbalance of surrogate PCR products, 12 ng whole-genome DNA library (balance library) was mixed with the four PCR samples in a final concentration of 1.5 ng/µl and sequenced in one lane. All the samples were subjected to pair-ended 150 bp deep sequencing on MGISEQ-2000 platform.

**Data analysis.** In order to evaluate the sequencing quality of amplicons and filter the low-quality sequencing data, Fastqc-0.11.3 and fastp-0.19.6[37] were used with default parameters for each sample. The clean sequencing reads of pair-ended segments were merged using FLASh-1.2.1[38] to obtain full-length reads. In order to obtain the amplified fragment reads of each surrogate reference sequence, BsmBI Linker was removed from the surrogate reference sequence. The BWA-MEM algorithm[39] of bwa was used for local alignment, and the reads of all samples were divided into 12,000 independent libraries. Due to the existence of sequencing or oligo-synthesis introduced errors, each library was then filtered. As SpCas9 mainly causes insertions and deletions, the length of surrogate sequence is expected to change from its original 37 bp. We adopt the following steps for data processing and filtering: (1) Obtain the sequence containing gRNA + scaffold fragment as dataset1. (2) Obtain the sequence containing GTTTGAAT in dataset1 as dataset2 (BsmBI linker fragments changed in orientation (GTTTGGAG− > GTTTGAAT)). (3) Extract the intermediate surrogate sequence from dataset2, which removed the length limit. In order to eliminate the interference of background noise before analyzing editing efficiency, all mutations or indels found in WT HEK293T cells group were removed.

The total editing efficiency for each gRNA was calculated according to the following formula:

$$\text{Total editing efficiency} = \frac{(\text{Num. reads with length} \neq 37 \text{ bp})}{(\text{Tot. num. of reads})}\% \quad (1)$$

The average fraction of indels from 30 bp deletion to 10 bp insertion was calculated according to the following formula:

$$\text{Average indels fraction} = \frac{(\text{Num. reads with length range}[7, 47] \text{ bp})}{(\text{Tot. num. of reads of 12K library})}\% \quad (2)$$

**Data collection and preprocessing for machine learning.** The 12 K dataset was preprocessed by removing gRNAs supported by less than 200 reads and by intersecting the datasets of gRNAs with efficiencies measured at day 8 ($N = 10,655$) and day 10 ($N = 10,933$), thus retaining data for 10,592 gRNAs. For training, efficiencies measured at day 8 and day 10, positively correlated (Pearson's $r = 0.91$), were averaged. The following additional datasets were downloaded: Kim (2019–2020)[7,9]; Wang (2019)[8]; Xu (2015)[17,21], Chari (2015, 293 T cells)[20]; and Hart (2015) Hct1162lib1Avg[18] as collected by Haeussler et al.[40]; Doench (2014–2016) from the public repository of the Azimuth project[14,19]. For the dataset

by Doench et al. (2014) only data from human cells was used, while for the later dataset (2016) we filtered for the genes CCDC101, CUL3, HPRT1, MED12, NF1, NF2, TADA1, TADA2B, as previously recommended[14,40], and excluded gRNAs marked for low early time point (ETP). The Wang (2019) dataset was filtered from gRNAs for which no context was defined in the corresponding study[8]. Based on the method used to evaluate gRNA activity, datasets were distinguished into two categories: loss of gene function studies, which comprises Xu (2015), Hart (2015), and Doench (2014–2016) and indel-based, including Kim (2019–2020), Wang (2019), Chari (2015) and this study.

The datasets were preprocessed by removing gRNAs matching one of the following criteria: (1) Not present in hg38 (except for exogenous constructs); (2) No match to the target gene based on GENCODE annotations (v 32); (3) High variance in efficiency between different experimental settings, above the threshold: upper quartile + 1.5× variance interquartile range; (4) Target gene with less than 10 designed gRNAs; (5) Related to a PAM different from 5'-NGG-3' (6) Expressed from a tRNA system; (7) Targeting the last 10% of the merged coding sequences (CDSs) annotated for a target gene (nonsense mediated decay or polymorphic pseudogene transcripts were excluded). Points 2, 4, and 7 were applicable only in the case of loss of function studies. The Kim (2019–2020) datasets were further processed by averaging duplicated 30mer gRNA + context entries (avg. difference between max. and min. indel frequency of replicates = 8.6 and 6.7 in the studies of 2019 and 2020, respectively). Efficiency values not reported as indel frequencies were ranked-normalized with the SciPy rankdata function[41] and normalized efficiencies were averaged between experimental conditions.

After preprocessing, each dataset contained the following number of unique 30mer, gRNA + context sequences: Kim (2019): 13,359; Kim (2020): 8742; Xu (2015): 971; Chari (2015): 1,224; Hart (2015): 4001; Doench (2014): 781; Doench (2016): 2145; Wang (2019): 55,022; this study: 10,592. See Supplementary Table 2 for more details about filtered data. Ours and Kim (2019) datasets were combined by building a linear regression model on overlapping elements (49 pairs) and applying it to scale gRNA efficiencies from our study to those of Kim et al. (2019). Efficiencies were averaged for overlapping 30mers. The merged dataset consisted of 23,902 gRNA + context sequences (30mers).

**Generation of gRNA and target DNA features.** Features were extracted from a 30mer DNA sequence composed by the target DNA protospacer (20 nt) and the following flanking regions: 4 nt upstream, 3 nt PAM, and 3 nt downstream from the PAM. Position-specific single and di-nucleotides were one-hot encoded, binarizing the presence/absence of a certain nucleotide with the values 0 (absent) or 1 (present). They were denoted as N_X, with N in the set [A,T,G,C] and X being the position on the 30mer. Nucleotides surrounding the "GG" Cas9 binding site were also binarized and denoted as NGGX_YZ, where Y and Z are the nucleotides upstream and downstream from the motif. Sliding windows of 1 and 2 nt were used to count the occurrences of each single and dinucleotide in the 30mer sequences. These position-independent features were labeled by the nucleotide or dinucleotide they account for. The GC content was obtained as the sum of Gs and Cs in the protospacer sequence. The melting temperatures were computed with the Biopython 1.77 Tm_staluc method[42] for three nonoverlapping segments of the protospacer, at positions 3–7, 8–15, and 16–20, referred to as MT_[S,E], where S and E are the start and end positions of the segment. The spacer folding free energy of ensemble and the $\Delta G_B$ RNA–DNA binding energy were computed using the energy function in the CRISPRoff pipeline 1.1.1[27], provided with RNAfold 2.2.5[42].

**Generation of dataset partitions.** The datasets used for training were divided into partitions of approximately equal size (±1 gRNA) accounting for data similarity, to assign highly similar gRNAs to the same partition. This was implemented as follows: (1) we computed the pairwise Hamming distance between all gRNAs based on their on-hot encoded 30mer sequences (gRNA + context) with the SciPy *pdist* function[41] (normalized distances from *pdist* were multiplied by the size of the one-hot encoded array (1 × 120)); (2) for each gRNA x we stored a list of all gRNAs with Hamming distance ≤ 8 in the one-hot space, which corresponds to a sequence difference ≤ 4 nt; these were regarded as gRNAs "similar" to gRNA x; (3) gRNAs similar to at least one other gRNA in the dataset were the first to be distributed, randomly, in the partitions; whenever a gRNA x was assigned to a partition, all the gRNAs y, z... similar to it (and recursively those similar to y, z, …) were also added to the same partition; (4) once all similar gRNAs were exhausted the remaining gRNAs, not similar to any other, were split into three subsets based on their efficiency (inefficient: up to efficiency percentile 25 (25p), medium-efficient: from 25 to 75p, and highly efficient: above 75p) and the gRNAs in these three subsets were distributed to the partitions pseudo-randomly by assigning a balanced amount of inefficient, medium-efficient and highly efficient gRNAs to each of the partitions until they reached their predetermined size. To preserve gRNAs from the test set of Kim et al. (2019) in a single partition, used as internal independent test set to compare the performances of CRISPRon and DeepSpCas9, the gRNAs in the test set of Kim et al. (2019) were collected in an initial group, which was assigned to the partition destined for usage as internal independent test set prior any other data partitioning. Other gRNAs in the merged dataset similar to any of the gRNAs present in this initial group were added to it during the generations of the partitions, to maintain the internal test set fully independent.

**Test settings for the evaluation and comparison of models.** Test datasets (both internal and external) were made fully independent by removing all gRNAs highly similar to a gRNA in the training sets of any of the models being compared as follows: (1) the pairwise Hamming distance between the gRNAs in the test and training datasets was computed using the on-hot encoded 20 nt gRNA sequences with the SciPy *cdist* function[41] (normalized distances from *cdist* were multiplied by the size of the one-hot encoded array (1 × 80)); (2) gRNAs with Hamming distance ≤ 6 in the one-hot space, which corresponds to a sequence difference ≤ 3 nt, were removed. While for the generation of dataset partitions gRNA similarities were computed on 30mer gRNA + context sequences, the sole 20 nt gRNA spacers were employed during the processing of the test datasets because in the dataset of Wang et al. (2019) target contexts are highly different from those in other datasets for identical gRNAs. More restrictive thresholds of similarity (sequence difference ≤ 4 or 5) were also tested. No difference in the general performance of CRISPRon (v0 and v1) and in the comparison with other models were observed, and all of the significant improvements remained as such (Supplementary Data 2). Notably, the fluctuations in performances given by different similarity thresholds were both positive and negative.

**Gradient boosting regression trees (GBRTs) for features analysis.** Validation hyperparameters were chosen from the following screen: learning rate chosen from [0.08, 0.09, 0.1], maximum tree depth chosen from [3, 5, 7], minimum number of samples to generate a new split chosen from [5, 10, 15, 20], minimum number of samples to be present in a leaf node chosen from [5, 10, 15, 20], total number of trees in the model chosen from [400, 600, 800, 1000]. The validation of hyperparameters was made twice, the first time on five out of six partitions of the dataset, preserving the 6th partition as internal independent test set, and the second time on all six partitions. Selected hyperparameters are reported in Supplementary Table 3. During the validation, each GBRT was initialized five times with different seeds and the best model of the 5 was chosen for each fold/validation set. Predictions were computed by averaging the output of the best GBRTs chosen for each fold. When comparing multiple predictors, independent test datasets were cleaned from gRNAs with ≤3 nt difference on the 20 nt sequence of a gRNA in the training set of any compared predictor.

**The CRISPRon deep learning model.** The training of our deep learning models uses the Keras/Tensorflow 2.2.0[43] neural network framework with Python 3.8.3. Our strategy takes outset in the deep learning strategies by Wang et al.[8] and Kim et al. (2019, 2020)[7,9]. We employed a one-hot encoding of the input sequence (30mer gRNA + context), which was fed into a number of 3, 5, and 7 sized filters acting directly on the one-hot encoded sequence. The convolutions, which are the outputs of the filters, were flattened and fed into two sequential fully connected layers before giving the gRNA efficiency as the final output (for the full model layout see Supplementary Fig. 13). The number of weights and the layout of the convolutions as well as those of the two final fully connected layers are identical to the architecture used in Kim et al. 2019 and since the hyperparameters and layout of their model was substantially interrogated, we have not attempted further optimizations of this part of our model for CRISPRon. However, the inclusion of an important biological parameter in the deep learning framework was optimized as detailed below.

The partitioning of the data in to 6 subsets used for the GBRT were reused in the training of the deep learning models (see Supplementary Table 1). As in the regular machine learning above, the deep learning models were initially trained on 5-fold cross-validation with the 6th partition set aside for internal independent testing. Each training in the 5-fold cross-validation was repeated 10 times using random seeds and the best model of the 10 was chosen for each fold/validation set. The final output is the average of the output of the best models chosen for each fold. Finally, the process was repeated using all six data partitions for 6-fold cross-validation without an internal independent test set.

The most important biological parameters obtained from the GBRT model Gini and SHAP analysis were $\Delta G_B$, the GC content of the 30mer and the folding energy of the spacer gRNA. Of these, $\Delta G_B$ was a far better representative of the on-target efficiency and we therefore decided to include $\Delta G_B$ in our deep learning model[27]. Direct inclusion of $\Delta G_B$ along-side the convolutions led to an improvement of the mean square error (MSE) from 143.15 to 141.76 on the average of the 5-fold cross-validations of the combined dataset from our study and Kim et al. (2019) (see Supplementary Table 4 and Supplementary Figs. 13-14 for the model layouts, Supplementary Table 5 for the results). Collecting the convolutions in a separate fully connected layer before combining the fully connected layer with $\Delta G_B$ led to a further improvement of the average MSE on the 5-fold cross-validation from 144.73 with three fully connected layers but without $\Delta G_B$ to 140.83 with $\Delta G_B$ (see Supplementary Table 4 and Supplementary Figs. 15-16 for the model layouts, Supplementary Table 5 for the results). The model with convolutions collected in a fully connected layer before combination with $\Delta G_B$ thus became our final CRISPRon model as outlined in Fig. 2a with details in Supplementary Fig. 16. This model was trained on the combined dataset from our study and Kim et al. (2019) dataset, split in six partitions, using 6-fold cross-validation. The final CRISPRon-v1.0 output is the average output of the best models obtained from each of the six validation sets after 10 repetitions (see Supplementary Table 6).

All the models were trained and evaluated using the MSE and the training was performed in epochs, where the weights were updated after each batch of 500 examples. The training was stopped when the performance on the validation set did not improve for 100 consecutive epochs and the best performing model by MSE on the validation set was kept. In effect, the training typically ran for 500–1500 epochs in total. The introduction of $\Delta G_B$ in the model changed the convergence behavior and we therefore screened for optimal learning rates testing learning as follows. We trained deep learning models on the LK-5 datasets using the layouts with only two fully connected layers and direct inclusion of $\Delta G_B$ and tested learning rates of 0.001, 0.0005, 0.0001, and 0.00005 in ten repetitions on each of the 5-fold validation sets (Supplementary Table 7). The optimal learning rate was 0.0001 using ADAM optimization and as above using a batch size of 500. The hyperparameters were used in the further training of the final CRISPRon deep learning model, which includes an extra fully connected layer for collection of the convolutions prior to the inclusion of $\Delta G_B$.

**Reporting summary**. Further information on research design is available in the Nature Research Reporting Summary linked to this article.

## Data availability

High-throughput sequencing data have been deposited to the China National GeneBank (accession number CNP0001031) and the GEO repository (accession number GSE173708). The gRNA efficiency data are provided in Supplementary Data 1. The Drugable gene database can be accessed to the link http://dgidb.org. The lentivirus vector used for cloning surrogate oligonucleotides is made available through Addgene (Plasmid #170459). Source data are provided with this paper.

## Code availability

CRISPRon website and source via: https://rth.dk/resources/crispr/ and on https://github.com/RTH-tools/crispron[22].

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

## Acknowledgements

This project was partially supported by the Sanming Project of Medicine in Shenzhen (SZSM201612074, to L.B. and Y.L.), Qingdao-Europe Advanced Institute for Life Sciences Grant (Y.L.), Guangdong Provincial Key Laboratory of Genome Read and Write (No. 2017B030301011 to X.X.), Guangdong Provincial Academician Workstation of BGI Synthetic Genomics (No. 2017B090904014 to X.X.), the Innovation Fund Denmark (4108-00008B, 4096-00001B to J.G.) and the Danish Research Council (9041-00317B to J.G.), Danish Research Council (9041-00317B to Y.L.), European Union's Horizon 2020 research and innovation program under grant agreement No 899417 (Y.L.), the Lundbeck Foundation (R219–2016-1375 to L.L.), the DFF Sapere Aude Starting grant (8048-00072 A to L.L.), and the National Human Genome Research Institute of the National Institutes of Health (RM1HG008525 to G.C.). We thank the China National GeneBank for the support of executing the project under the framework of Genome Read and Write.

## Author contributions

Y.L. and J.G. conceived the idea. L.B., G.M.C., C.A., L.L., and Y.L. supervised and coordinated the experimental part of the study. J.G. supervised and coordinated the computational part of the study. X.X., K.Q., X.L., and X.P. performed most of the experimental work. G.I.C., C.A., X.X., and X.P. analyzed the data. G.I.C., C.A., and J.G. developed the machine learning methods. G.I.C. and C.A. implemented the deep learning methods and webserver. All authors have contributed to the execution of the experiments and studies. X.X., G.I.C., L.L., C.A., J.G., and Y.L. drafted the manuscript. All authors discussed the results and contributed to the final manuscript.

## Competing interests

The authors declare no competing interests.
