## [Peer Review File · Nature Communications]

Reviewers' Comments:

Reviewer #1:

Remarks to the Author:

This manuscript by Xiang, Qu, Liang, Pan et al. describes an assay and resulting datasets for quantification of Cas9-nuclease, ABE, and CBE genome editing outcomes for 12,000 gRNAs in human HEK293T cells. The authors perform analysis to measure and predict editing efficiency and editing outcomes using the data they have collected. While there is an abundance of data and analysis, ultimately it is difficult to determine how meaningful much of it is because of technical flaws in the design, execution, and follow-through (or lack thereof) of the experiments. Because of these flaws, this work should not be published in its current form, as the veracity of its conclusions cannot be trusted. Below, I list a number of major and minor points that should be addressed for this work to provide meaningful and trustworthy information to the genome editing community.

Major points

1. TRAP-seq describes a paired gRNA/target site approach that has now been published in several papers from 2018 onwards (Shen et al, Allen et al to name a few). This established technique should not get a new name. The authors should remove all mentions of TRAP-seq and just call this approach a gRNA and target library approach as others have before unless they can properly justify why this is a qualitative advance over these previous methods.
2. In fact, previous versions of gRNA and target site approaches (Shen et al, Allen et al) have used Gibson Assembly cloning to lengthen the target site and surrounding sequence to 55-79-nt (this work uses a 37-nt target site) and have shown that this broadened target site more accurately captures Cas9 outcomes. The authors should acknowledge they have used a strictly less faithful approach to measuring Cas9 outcomes when others have published superior approaches 1-2 years ago.
3. The appropriate way of determining if a computational method provides accurate predictions at native sites is to test a large collection of native sites in their cell line and compare predictions to data. The authors do not do this for any of their approaches. In all cases where they build algorithms based on this synthetic data that they claim should translate to native loci (gRNA efficiency prediction, ABE and CBE outcome prediction), they must use a large collection of native sites to test the accuracy of their algorithm. Otherwise, the algorithm only works on synthetic data and is not useful to others in the field. This is a big deal, as it is common for algorithms to overfit toward data from a particular system and not generalize outside of this system.
4. There are several issues with the gRNA efficiency prediction:
 - a. The authors predict editing efficiency through their BRR model and compare the results to actual editing efficiencies by use of spearman correlation coefficient to determine accuracy. If there was some discrepancy in presence of features (2,483) applying to few (or excessive) samples during training you run into the issue of those cases taking priority when it comes to evaluation. The issue I see here is that these samples' predictions may be ordinally similar in their BRR but may be off in terms of the actual difference in predicted editing efficiency vs observed. Different and more traditional measures of logistic regression performance such as standard error or MSE would better capture these differences. It's tough to say that a predictive model is performing well without evidence of robustness of predictions, and accuracy among all parts of the distribution.
 - b. The comparison of GNL-Scorer to other algorithms is not well described. Were these other algorithms trained on the data from the 80% training data from this current work as well? If not, then this presents an unfair comparison of algorithmic performance. Since it seems that DeepCas9 and Azimuth2.0 display the most comparable generalized performance, it would be most fair to compare performance of each of these algorithms trained on the same 80% training data from this work to determine the relative performance of these models trained on an identical training set (and evaluated using MSE as noted above in addition to Spearman).
 - c. gRNA efficiency is not smoothly distributed. Most gRNAs appear to have high efficiency, and then there is a long tail of lower efficiency among a small set of gRNAs. This distribution favors a binning method over a continuous scoring method as being most useful in practice. The authors should predict bins (quartiles, quintiles, or the like) and ask how well each gRNA prediction

algorithm predicts the gRNA efficiency bin. This may provide a more useful metric of utility of each algorithm.

5. Relatedly, Figure S2 seems to display a spuriously high r^2 because of high correlation at the tail ends. MSE is a more appropriate way of calculating the similarity between their predictions and actual data.

6. If there was substantial editing prior to Dox addition, then this editing could have occurred before lentivirus integrated into the genome. Thus, the data from this work does not necessarily represent genome-integrated constructs. The authors should clearly acknowledge this caveat. Ideally they would repeat all experiments that showed substantial editing in the absence of Dox. But if not, this emphasizes again the need for independent validation of individual native genomic sites.

7. Targeting native genes could cause biases if the native targets are essential or inhibit growth, as mutations in the native counterparts of these sites could impair cell growth. The authors should compare Cas9-nuclease and CBE stop-gain mutation efficiency in essential genes (Cancer DepMap) represented in their library to determine whether their results are confounded by native gene essentiality, and if so they should discuss the implications for their work.

8. Because the authors do not design their library randomly for ABE and CBE testing, it is possible that they may not have equivalent representation of 3-mer sequences centered on editable bases, which would cause blind spots in their model. They should provide a supplementary figure or table that describes the representation of each 3-mer with an A or C in the center.

9. The authors should compare their ABE and CBE predictive models to the newly published BE-Hive model (Arbab, Shen et al Cell). While this did come out after I received this paper to review, it is too relevant for them not to compare algorithmic performance.

Minor points

1. Figure 2e shows a discrepancy of 1bp insertions among four nucleotides, and insertions are highly dependent on the N17 nucleotide (fig 2f). Methods mentions that features in the BRR include 604 one-hot encodings of nucleotide(s), leading me to think that among their gRNAs there may have been an overabundance of certain features, as in N17 nucleotide, given the findings shown in 2e and 2f. The authors should normalize Fig. 2e based on N17 base identity.

2. The most significant features by SHAP analysis seem to be taking priority among these predictions without much regard for minor features that don't influence ordinality of the results. Thus, it would be worth asking whether their model changes if a different accuracy metric is used other than r . As an example, the authors show that the motif percentages in N5-N7 are over represented in both high and low efficient guides (fig 4c). If this is going into the model as a feature, it might support the idea that a lot of these guides are taking on this feature weight that aligns with a high spearman correlation, but might not align with other measures of regression accuracy. Looking at the SHAP values, some of the most relevant features are related to GC count or other counts of G and C nucleotides. If CAG is showing up 35% of the time in positions N5-N7 in the extreme cases (less than 1% efficiency and over 20% efficiency), the adjacent nucleotides will give rise to a GC motif more often even if it's by chance (25% of the time). Since GC content is a big driver of the prediction according to SHAP, this could be largely influenced by the aforementioned GC issue that will arise in the extreme cases, where ~35% of the extreme cases are high in GC content, in training these examples will influence the weight of these parameters, and predictions on the gRNAs with this feature that are not in the high or low end of editing efficiency may suffer from the increased weight of this feature being bolstered. This would lead to what I explained previously in terms of things correlating in terms of spearman correlation, due to the same weight being added to their prediction, but may not be indicative of proximity to actual distance to the observed frequency. The authors should explain or account for this possible source of bias.

3. Observing the distribution of r values in fig S17 may indicate few points were going into this comparison, and the relatively uniform distribution among some of the figures doesn't support the claim that editing outcomes were correlated without providing levels of significance.

4. Comparisons to inDelphi predictive outcomes are not well described in terms of what exactly is being compared.

5. The analysis of Cas9 repair outcome distributions does not add any novelty to existing published work. As such, it should be considered whether this should be moved to the supplement.

6. Typos

a. Page 15 figure legend g misses the word THAT "between gRNAs THAT have"

b. Page 15 figure d misses the "n" values marked as in figure g

Reviewer #2:

Remarks to the Author:

The manuscript by Xiang and colleagues describes a high-throughput method for assessing editing outcomes at 12,000 artificial sites using either Spy Cas, an adenine base editor, or a cytosine base editor. Using this approach, the authors were able to assess the efficiency at the edited sites as well as to deduce properties of each target site that could improve or hinder the likelihood of indel formation or base conversion. The data were also used to train a machine learning-based prediction tool that can be used for prediction of outcomes for additional sites.

Critique of the manuscript:

The claims made in lines 121-124 are not well-supported. If more data supporting these claims is not provided, then this section should be substantially re-phrased. The data presented in Figure S2 are so far removed from the claims made that it casts severe doubt on the entire manuscript.

The comparison of CBE outcomes at TRAP vs. endogenous sites (Fig. S2e) clearly demonstrates a substantial difference in editing efficiencies: 28 or 20% C>T conversion in the cassette data, but only 12 or 9% (respectively) in the genome. This suggests that the editing efficiencies at the two loci may substantially differ, even if the relative efficiencies are preserved (e.g. the first site is edited ~40% more efficiently than the second site). The ABE results suggest a good correlation, but there is inadequate data here to make any conclusions.

However, the correlation values cited in the paper – "($r^2 = 0.96 - 0.99$)" – are founded on a deeply flawed approach. Plotting TRAP site vs. genomic site correlations of all the positions in the relevant window (as is done in panels S2d+f), is an inexplicable choice that floods the analysis with irrelevant information. Plotting of all the bases that are not expected to change causes clusters of irrelevant data near "0%" and "100%" (the latter is for G positions in S2d, for T positions in S2f) which dominate the "best fit" line and will ensure a very high r^2 value, regardless of the correlations we are actually interested in. Namely, the C>T sites in S2f, which are clear outliers, yet this critical finding is obscured by the pointless r^2 value and is treated as non-existent by the claims made in the paper (e.g. "The results validated that the CRISPR editing efficiency and outcomes from the surrogate sites were closely correlated").

In reference 27, the authors were inspecting indel formation, and their correlations (between endogenous and exogenous sites) are not particularly compelling: 0.65 to 0.82 with adjusting; $r = 0.52$ to 0.76 without adjusting. The portions of reference 29 that seem to be relevant (presumably Figure 1h and the related results) are not statistically powered (e.g. there are not replicates) and comparison of a single T7E1 assay gel to Sanger analysis of 100 clones is an "apples to oranges" situation with unclear informational value, especially regarding indel profiles. These two references do not support the claim that "a surrogate target site can faithfully recapitulate the endogenous editing efficiency and indel profile". At best, the referenced work seems to demonstrate that surrogate sites can approximate endogenous editing events.

The authors must address this glaring issue before this manuscript can be considered for publication. The best way to address this issue would be to perform deep sequencing of the same type (and analysis pipeline) used for the 12,000 sites that are the main focus of the paper. Using Sanger/ICE analysis for benchmarking of two locus pairs (ABE or CBE, TRAP site vs. genomic site)

is not sufficient to provide compelling evidence that surrogate loci recapitulate editing outcomes. In addition to an appropriate method of detection/analysis, the authors are encouraged to perform multiple technical replicates per site as well as checking multiple locus pairs. The previous choice of CBE site (TYPM) was appropriate: it allowed assessment of both the overall editing efficiency as well as relative efficiency at sites within the base editor's active window. More similar CBE loci should be examined, as well as multiple ABE loci with similar properties. The ABE locus (INHBC) allows comparison of overall efficiency but not relative efficiency at sites within the base editor's active window.

Note that the above critiques apply to any time Fig. S2 is referenced (e.g. line 293).

Day 2 editing data seem to represent anomalous outcomes that may not represent typical genome editing outcomes and may instead reflect some sort of experimental artifact specific to the early time-point. Perhaps this is because the lentivirus itself is still present at Day 2. In particular, the Day 2 data in Fig. 1e (the pie chart) reveal a pattern where "other" outcomes predominate: ~70% of outcomes are "other" on Day 2 but <20% on Day 8 and Day 10. Furthermore, the fact that Day 2 outcomes have a very poor correlation with inDelphi predictions (Fig. S17) suggests that something anomalous is happening. The authors must discuss these "other" outcomes in greater detail; what is their nature, and why might they appear so frequently in the Day 2 results? The Fig 1e caption suggests that these could be wild-type reads, which would be confusing if true, because there are no apparent wild-type (0% editing) reads in the Day 2 data of Fig. 1d. Please clarify what is responsible for the unexpected Day 2 results of Fig. 2e and Fig. S17. The current statements at lines 224-227 could also be updated; it seems unlikely that this shift can be attributed to "experimental conditions". If there is any precedent for an observation like this (dramatic change in indel patterns), it should be discussed. I am not aware of any such precedent.

The above issues must be addressed in a revised manuscript before it can be seriously considered for publication.

It may be helpful to talk about the relationship between the data in Fig. 2f and 2g. Perhaps it would be reasonable to consider a model where most indel repairs are driven by insertion of a base matching that at position N17, but in the case of a "G" at N17, deletion is favored. This may explain the weaker influence of "G" at N17 on insertions, as we see in Fig. 2f.

Consider using consistent colors in Fig. 2e+f for A/C/G/T. It is confusing that they are currently different. Supplemental figures would ideally also follow a single, consistent theme, so this would call for an update of the colors in Fig. 2e.

Reconsider the name "TRAPseq" since an essentially identical term has already been established, e.g. "Translating Ribosome Affinity Purification (TRAP) Followed by RNA Sequencing Technology (TRAP-SEQ)"

Figure S5: Although many of the bars have error bars that are nearly invisible, analysis of the vector art demonstrates that at least one of these bars (library 2, 10 μ L) has no error bars at all. Please make note of this, if it is fact an $n = 1$ condition.

Supplementary Tables: Perhaps I missed this information, but it seems that there are no legends provided for the supplementary tables. If that is the case, it should be rectified. It is not clear what the values represent in Tables S3 & S4.

Throughout the text, it would be extremely valuable to replace all instances of "guide sequence" with "spacer sequence". The "guide sequence" refers to the full RNA that Cas9 binds to, either in sgRNA or dgRNA format. The 5' portion of the crRNA or sgRNA that determines what DNA sequence is being targeted is referred to as the "spacer" and it seems that this sequence is the focus of the secondary structure predictions in this work. Furthermore, the materials/methods

section on this topic (section "vi" specifically) is currently incomprehensible regarding the secondary structure prediction that was performed. Please make this clear and provide adequate detail so the experimental procedures can be understood and potentially repeated.

Regarding the interpretation of Figure 2a+b, it seems that the claims are not fully supported. There is no mention of statistically significant differences between different clusters/bins of data (e.g. 30-40% vs. 40-50% in 2a), so it is unclear how/why claims are being made regarding the "optimal" ranges.

"Importantly, according to our knowledge, this is the first time that both ABE and CBE efficiencies are measured at such a large scale in cells."

I'm not sure if priority/novelty claims are allowed in this journal, but even if you are permitted to make such a claim, it may no longer be true due to the recent Arbab et al. publication.

Feedback on readability:

Fig. S38b Labels should read "stopped"

77 Should be "attractive tools".

80-81 Base editors are described as able to create "a [...] substitution" within the editing window, but in many cases there are multiple edits within the window. Consider revision to "substitutions".

82 Consider changing "Albeit" to "Even with"

83 Change to "urgent need for", change "DBS" to "DSB"

57-87 The entire introduction has excessive focus on Cas9. Many statements that refer to editors in general are currently making reference to Cas9. Instead, consider referring to "genome editors", "genome editing enzymes", or "RNA-guided endonucleases (RGENs)" to be more inclusive of relevant technologies.

89-91 For many scientists (especially those working with animals), the term "in vitro" is used to include tissue culture experiments. To distinguish between cell-free (biochemical/enzymatic) assays and tissue culture assays, you should use other terminology.

93-4 Consider revising "we lack" to "there is a deficit of". This will convey that we don't have enough data. "Lack" suggests that we have no data, and currently there is at least one other report that contains large-scale ABE/CBE data.

115 Consider revising to "of at most 170 bp"

115 "170 bp" is not consistent with 37bp". Use the former style (with a space) throughout, since all values should be separated from units by a space. Same issue on line 333 - be consistent.

131-2 Correct to "Figure not drawn to scale."

199-200 Consider revising to "we were capable of analyzing the likely consequences of all indels on protein translation". This change is important because sometimes the actual outcomes differ from predictions.

230 Should be "therefore"

234-235 Should this be "also see the CRISPR Atlas resource" or "also seen in the CRISPR Atlas resource"?

281 "GC content delta G energy" if this is meant to represent two distinct features, add a comma.

283-286 This sentence is hard to understand, especially regarding the "efficiency prediction..." clause, the "efficiency dataset" clause and their relationship to other clauses.

287 Consider revision to "has been deposited and is publicly available at GitHub"

290 Same suggestion as lines 93-4.

376 Should be "analyzed"

421 Should be "First, based on the optimized"

810 Should be "split" ("splitted" is not in use, typically)

906-907 "For the editing window is wider" - it seems that a word might be missing here. Check the entire sentence and make it more clear if possible.

Dear reviewers,

We would like to acknowledge the critical comments and constructive suggestions to improve our study and manuscript. In this revision, we have extensively revised our study, the analyses and the manuscript. A few major changes are highlighted here, while a more detailed point-by-point response to all the comments are provided below this letter. In brief, we have:

1. Made a validation of the TRAP-seq (in the revision called CRISPRTRAP-seq) in 16 endogenous sites and corresponding TRAP loci (Spearman's $R = 0.72$, p value = 0.0016, Figure S3 of the revision). In addition, we co-validate gRNA efficiency measured by CRISPRTRAP-seq with those measured by Kim et al. (2019) (number of gRNAs = 49) and Wang et al. (2019) (number of gRNAs = 207). Good correlations were obtained for both comparisons ($R = 0.67$). Results are showed in revised Figure 1g.
2. Made substantial changes to the structure, analyses and content as compared to the previous version. Instead of presenting all SpCas9, ABE and CBE dataset in one study, we now focus only on SpCas9 and present the CRISPRTRAP-seq approach for capturing on-target SpCas9 gRNA activity. Most importantly, we integrate our on-target SpCas9 gRNA efficiency data with published data from Kim et al. (2019) to develop a deep learning based on-target SpCas9 gRNA activity predictor, CRISPRon, which outperforms the most recent and cutting-edge popular predictors (Fig. 2, Figure S11).
3. Made an in-depth analysis of previous Cas9 on-target prediction methods and show that the high performance (~ 0.93 in Spearman) published by Kim et al. (2020) in *Nature Biotechnology* is inflated due to inclusion of data from non-targeting PAMs. The data are furthermore suboptimal in their representability relative to existing data in the field (Fig. S1, Supplementary Note 1).
4. Made an interactive webtool and a standalone software for on-target gRNA designed is developed based on CRISPRon (<https://rth.dk/resources/crispr/crispron/>). The CRISPRon is available online as stated in the abstract. The software has not yet been made online, but is enclosed here for the reviewing purposes and will be made online when the manuscript is being published.

On behalf of all authors,
Jan Gorodkin and Yonglun Luo

Reviewers' comments:

This manuscript by Xiang, Qu, Liang, Pan et al. describes an assay and resulting datasets for quantification of Cas9-nuclease, ABE, and CBE genome editing outcomes for 12,000 gRNAs in human HEK293T cells. The authors perform analysis to measure and predict editing efficiency and editing outcomes using the data they have collected. While there is an abundance of data and analysis, ultimately it is difficult to determine how meaningful much of it is because of technical flaws in the design, execution, and follow-through (or lack thereof) of the experiments. Because of these flaws, this work should not be published in its current form, as the veracity of its conclusions cannot be trusted. Below, I list a number of major and minor points that should be addressed for this work to provide meaningful and trustworthy information to the genome editing community.

Re: We appreciate the reviewer's constructive suggestions for the improvement of study, the manuscript and the ensuring that only meaningful and useful methods and information will be delivered to the genome editing community. While the in meantime, we do not totally agree with the reviewers on some suspected critics on the technical solidity and results. Despite that, we have thoroughly taken all reviewer's comments and suggestions into consideration and provide a substantially validated and revised manuscript. Please also noted that in this revision, we have decided to mainly focusing on presenting the CRISPRTRAP-seq approach, SpCas9 dataset generated, and most importantly, the development of the CRISPRon prediction model and tool. The ABE and CBE dataset are removed from the revision in order not to diverse the focus of the study. The ABE and CBE dataset will be presented in another study.

Major points

1. TRAP-seq describes a paired gRNA/target site approach that has now been published in several papers from 2018 onwards (Shen et al, Allen et al to name a few). This established technique should not get a new name. The authors should remove all mentions of TRAP-seq and just call this approach a gRNA and target library approach as others have before unless they can properly justify why this is a qualitative advance over these previous methods.

Re: As already acknowledged in our previous submission and in the revised studies, we have cleared stated that the CRISPRTRAP-seq is an optimized protocol based on Shen et al, Allen et al in 2018. Back in 2016, we have already reported in-cell surrogate sites (C-Check method) can faithfully capture CRISPR gRNA activity (Cell Mol Life Sci. 2016 Jul;73(13):2543-63.). While the reviewer is strongly against of using an acronym to described modified approach, we strongly feel that with an acronym to describe the approach is much better than as "a gRNA and target library". Besides, our approach has implemented several optimizations to broaden and streamline the generation of libraries (see below response to point 2). We have in the revision sanitized the description of the CRISPRTRAP-seq approach.

2. In fact, previous versions of gRNA and target site approaches (Shen et al, Allen et al) have used Gibson Assembly cloning to lengthen the target site and surrounding sequence to 55-79-nt (this work uses a 37-nt target site) and have shown that this broadened target site more accurately captures Cas9 outcomes. The authors should acknowledge they have used a strictly less faithful approach to measuring Cas9 outcomes when others have published superior approaches 1-2 years ago.

Re: The study by Shen et al. used 55-bp target sites and the study by Allen et al. used 79-nt target sites. While we acknowledge that longer surrogate sites preserve more surrounding sequence characteristics of the target sites, however none of these studies have demonstrated that 79-nt is better than 55-nt, and more faithful than 37-nt. While in the study of Wang et al. (Nature Communications volume 10, Article number: 4284 (2019)), using 23-nt target sites have been demonstrated to faithfully capture gRNA efficiency. Most current CRISPR gRNA activity predictors are based on 30-37 mere encompassing the protospacer sequences. The majority of CRISPR-induced indels are deletions less than 30bp and insertion of 1-2bp. While both Gibson Assembly and Golden Gate Assembly are two broadly used methods for vector cloning, our approach offer the CRISPR gene editing community with the Golden Gate Assembly strategy. We have introduced several improvements to the lentiviral vectors to streamline all experimental steps: from vector cloning (Golden Gate Assembly with blue and white selection), lentiviral packaging (functional titer quantification by evaluating EGFP positive cells), to selection of stably integrated cells (puromycin selection and EGFP monitoring). The vector system will be made available through the Addgene plasmid depository.

3. The appropriate way of determining if a computational method provides accurate predictions at native sites is to test a large collection of native sites in their cell line and compare predictions to data. The authors do not do this for any of their approaches. In all cases where they build algorithms based on this synthetic data that they claim should translate to native loci (gRNA efficiency prediction, ABE and CBE outcome prediction), they must use a large collection of native sites to test the accuracy of their algorithm. Otherwise, the algorithm only works on synthetic data and is not useful to others in the field. This is a big deal, as it is common for algorithms to overfit toward data from a particular system and not generalize outside of this system.

Re: We agree with the editor for this criticism. In the revision, we have provided three independent validations to consolidate the method. We have also performed systematic benchmarking analyses among different datasets, different prediction models, and most importantly, we developed the deep learning CRISPRon which outperformed all other models in 4 test datasets not overlapping with training data used for the development of these tools.

A

	TET2		indel	reads	freq.			
	protospacer	PAM						
Endogenous locus	TTC	AAGAACAGGAGCAGAAGTCA	CAACAAGCTT	CAGTTCACAGGGATATAAAAAATAGAAACCAAGATATGTCTGGTCAAC	I1	397777	46.51	
	TTC	AAGAACAGGAGCAGAAGTCA	CAACAAGCTT	CAGTTCACAGGGATATAAAAAATAGAAACCAAGATATGTCTGGTCAAC	D8	46887	5.48	
	TTC	AAGAACAGGAGCAGAAGTCA	CAACAAGCTT	CAGTTCACAGGGATATAAAAAATAGAAACCAAGATATGTCTGGTCAAC	I1	37436	4.37	
	TTC	AAGAACAGGAGCAGAAGTCA	CAACAAGCTT	CAGTTCACAGGGATATAAAAAATAGAAACCAAGATATGTCTGGTCAAC	D3	21473	2.51	
	TTC	AAGAACAGGAGCAGAAGTCA	CAACAAGCTT	CAGTTCACAGGGATATAAAAAATAGAAACCAAGATATGTCTGGTCAAC	I1	17925	2.09	
	TTC	AAGAACAGGAGCAGAAGTCA	CAACAAGCTT	CAGTTCACAGGGATATAAAAAATAGAAACCAAGATATGTCTGGTCAAC	D15	7504	0.87	
	TTC	AAGAACAGGAGCAGAAGTCA	CAACAAGCTT	CAGTTCACAGGGATATAAAAAATAGAAACCAAGATATGTCTGGTCAAC	I1	5786	0.67	
	TTC	AAGAACAGGAGCAGAAGTCA	CAACAAGCTT	CAGTTCACAGGGATATAAAAAATAGAAACCAAGATATGTCTGGTCAAC	I1	4632	0.54	
	TTC	AAGAACAGGAGCAGAAGTCA	CAACAAGCTT	CAGTTCACAGGGATATAAAAAATAGAAACCAAGATATGTCTGGTCAAC	I1	4455	0.52	
	TTC	AAGAACAGGAGCAGAAGTCA	CAACAAGCTT	CAGTTCACAGGGATATAAAAAATAGAAACCAAGATATGTCTGGTCAAC	D12	3221	0.37	
			protospacer	PAM				
	TRAP locus	GC	AAGTCA	CAACAAGCTT	CAGTTCACAGGGATAT	I1	558505	53.28
GC		AAGTCA	CAACAAGCTT	CAGTTCACAGGGATAT	D8	57877	5.52	
GC		AAGTCA	CAACAAGCTT	CAGTTCACAGGGATAT	D3	32657	3.09	
GC		AAGTCA	CAACAAGCTT	CAGTTCACAGGGATAT	I1	21629	2.06	
GC		AAGTCA	CAACAAGCTT	CAGTTCACAGGGATAT	D15	12177	1.16	
GC		AAGTCA	CAACAAGCTT	CAGTTCACAGGGATAT	D15	11370	1.08	
GC		AAGTCA	CAACAAGCTT	CAGTTCACAGGGATAT	D8	5009	0.47	
GC		AAGTCA	CAACAAGCTT	CAGTTCACAGGGATAT	D16	4966	0.47	
GC		AAGTCA	CAACAAGCTT	CAGTTCACAGGGATAT	D5	4500	0.42	
GC		AAGTCA	CAACAAGCTT	CAGTTCACAGGGATAT	I1	4373	0.41	
		protospacer	PAM					
Endogenous locus		TCT	GTGTCAGCTCCAGAAGTGC	TGGCCAGAAACCTTACAGCAAGC	TGTGGATTGCTGGTCCATCGGCGTCATCACCTAC	I1	239843	46.38
	TCT	GTGTCAGCTCCAGAAGTGC	TGGCCAGAAACCTTACAGCAAGC	TGTGGATTGCTGGTCCATCGGCGTCATCACCTAC	D3	15269	2.95	
	TCT	GTGTCAGCTCCAGAAGTGC	TGGCCAGAAACCTTACAGCAAGC	TGTGGATTGCTGGTCCATCGGCGTCATCACCTAC	D2	9108	1.76	
	TCT	GTGTCAGCTCCAGAAGTGC	TGGCCAGAAACCTTACAGCAAGC	TGTGGATTGCTGGTCCATCGGCGTCATCACCTAC	I2	8831	1.70	
	TCT	GTGTCAGCTCCAGAAGTGC	TGGCCAGAAACCTTACAGCAAGC	TGTGGATTGCTGGTCCATCGGCGTCATCACCTAC	D11	6472	1.25	
	TCT	GTGTCAGCTCCAGAAGTGC	TGGCCAGAAACCTTACAGCAAGC	TGTGGATTGCTGGTCCATCGGCGTCATCACCTAC	D8	3647	0.70	
	TCT	GTGTCAGCTCCAGAAGTGC	TGGCCAGAAACCTTACAGCAAGC	TGTGGATTGCTGGTCCATCGGCGTCATCACCTAC	D20	3439	0.66	
	TCT	GTGTCAGCTCCAGAAGTGC	TGGCCAGAAACCTTACAGCAAGC	TGTGGATTGCTGGTCCATCGGCGTCATCACCTAC	D6	3388	0.65	
	TCT	GTGTCAGCTCCAGAAGTGC	TGGCCAGAAACCTTACAGCAAGC	TGTGGATTGCTGGTCCATCGGCGTCATCACCTAC	D42	3216	0.62	
	TCT	GTGTCAGCTCCAGAAGTGC	TGGCCAGAAACCTTACAGCAAGC	TGTGGATTGCTGGTCCATCGGCGTCATCACCTAC	D2	3192	0.61	
			protospacer	PAM				
	TRAP locus	CAG	AAGTGC	TGGCCAGAAACCTTACAGCAAGC	TGTGGATTGCTGGTCCATCGGCGTCATCACCTAC	I1	222904	44.52
CAG		AAGTGC	TGGCCAGAAACCTTACAGCAAGC	TGTGGATTGCTGGTCCATCGGCGTCATCACCTAC	D16	18499	3.69	
CAG		AAGTGC	TGGCCAGAAACCTTACAGCAAGC	TGTGGATTGCTGGTCCATCGGCGTCATCACCTAC	D3	11883	2.38	
CAG		AAGTGC	TGGCCAGAAACCTTACAGCAAGC	TGTGGATTGCTGGTCCATCGGCGTCATCACCTAC	I2	10265	2.01	
CAG		AAGTGC	TGGCCAGAAACCTTACAGCAAGC	TGTGGATTGCTGGTCCATCGGCGTCATCACCTAC	D21	10279	2.01	
CAG		AAGTGC	TGGCCAGAAACCTTACAGCAAGC	TGTGGATTGCTGGTCCATCGGCGTCATCACCTAC	D11	8761	1.73	
CAG		AAGTGC	TGGCCAGAAACCTTACAGCAAGC	TGTGGATTGCTGGTCCATCGGCGTCATCACCTAC	D3	8640	1.73	
CAG		AAGTGC	TGGCCAGAAACCTTACAGCAAGC	TGTGGATTGCTGGTCCATCGGCGTCATCACCTAC	D8	7245	1.44	
CAG		AAGTGC	TGGCCAGAAACCTTACAGCAAGC	TGTGGATTGCTGGTCCATCGGCGTCATCACCTAC	D8	6652	1.33	
CAG		AAGTGC	TGGCCAGAAACCTTACAGCAAGC	TGTGGATTGCTGGTCCATCGGCGTCATCACCTAC	D2	5506	1.10	
		protospacer	PAM					

B

We performed targeted deep sequencing to analyze the indel frequency at endogenous sites and corresponding TRAP loci and achieve good corrections for these sites tested. (Figure S3)

We performed independent gRNA activity validation for overlapping gRNAs quantified by our study, by Wang et al. (2019), and by Kim et al. (2019). Our results showed that gRNA activities captured by our approach is well correlating with the other two studies (Fig. 1g).

4. There are several issues with the gRNA efficiency prediction:

a. The authors predict editing efficiency through their BRR model and compare the results to actual editing efficiencies by use of spearman correlation coefficient to determine accuracy. If there was some discrepancy in presence of features (2,483) applying to few (or excessive) samples during training you run into the issue of those cases taking priority when it comes to evaluation. The issue I see here is that these samples' predictions may be ordinaly similar in their BRR but may be off in terms of the actual difference in predicted editing efficiency vs observed. Different and more traditional measures of logistic regression performance such as standard error or MSE would better capture these differences. It's tough to say that a predictive model is performing well without evidence of robustness of predictions, and accuracy among all parts of the distribution.

b. The comparison of GNL-Scorer to other algorithms is not well described. Were these other algorithms trained on the data from the 80% training data from this current work as well? If not, then this presents an unfair comparison of algorithmic performance. Since it seems that DeepCas9 and Azimuth2.0 display the most comparable generalized performance, it would be most fair to compare performance of each of these algorithms trained on the same 80% training data from this work to determine the relative performance of these models trained on an identical training set (and evaluated using MSE as noted above in addition to Spearman).

c. gRNA efficiency is not smoothly distributed. Most gRNAs appear to have high efficiency, and then there is a long tail of lower efficiency among a small set of gRNAs. This distribution favors a binning method over a continuous scoring method as being most useful in practice. The authors should predict bins (quartiles, quintiles, or the like) and ask how well each gRNA prediction algorithm predicts the gRNA efficiency bin. This may provide a more useful metric of utility of each algorithm.

5. Relatedly, Figure S2 seems to display a spuriously high r^2 because of high

correlation at the tail ends. MSE is a more appropriate way of calculating the similarity between their predictions and actual data.

Re (points 4-5): We appreciate all the comments and critics from the reviewer to the previous GNL model. To ensure the gRNA efficiency prediction model presented is of robustness, accuracy and advance, we have collaborated with Jan Gorodkin's team to completely re-assess our data, remove gRNA targeting essential genes that could cause biases and remove low quantity sequencing sites. We additionally processed the selected public datasets to ensure high quality in both training and testing. In regard to this, in Supplementary Note 1 we show how the performance and benchmark of previously published predictors (and in particular of the one presented in Kim *et al. Nature Biotechnology* (2020)) have been biased by unfortunate choices regarding training and test data (Fig. S1c).

Figure S1. Evaluation of recent datasets and models of CRISPR gRNA efficiency. a. Comparison between the validation performances of different ML models on Kim *et al.* (2019) dataset. Black and red dots correspond to the Spearman correlation between experimentally measured and predicted indel frequencies obtained from one of 10 cross-validations (black), which are summarized as one box plot for each model architecture, or to the internal independent test set (red), computed by averaging the predictions obtained from the 10 trained models. The highest Steiger's test P-value obtained from 10 comparisons, one for each validation set is illustrated on top of the box plots. The suffix 'K' is appended to the model's identifier to designate trees validated with the set of parameters from Kim *et al.* b. Generalization performances of DeepSpCas9 and GBRTs, evaluated as Spearman correlation between experimentally measured and predicted indel frequencies. Statistical significance is

computed between DeepSpCas9 and GBRTs: Steiger's test *P < 0.05, **P < 0.01, ***P < 0.001, NS = not significant. c. DeepSpCas9variants performances decrease after removing non-canonical PAMs. On the X axis PAMs are sorted by median efficiency (left Y axis). Prediction performances (right Y axis) computed for DeepSpCas9variants on the full test set and after removing one PAM at a time, from left to right. d. Skewed distribution of indel frequencies for gRNAs in the dataset of Wang et al. (2019) compared to the Kim et al. (2019) and CRISPRTRAP-seq.

Based on a more robust training dataset we developed CRISPRon, a deep learning-based model that predicts gRNA efficiency. We also train a gradient boosting regression tree on the same data and employ it to analyze the importance of features, similarly to Kim *et al. Nature Biotechnology* (2020).

We address the main points raised about the previous version of the work below:

- (a)** The Spearman correlation between predicted and experimentally verified efficiencies has been previously employed by several studies on gRNA efficiency prediction, including Doench *et al. Nature Biotechnology* (2016), Kim *et al. Science advances* (2019), Wang *et al. Nature Communications* (2019), Kim *et al. Nature Biotechnology* (2020) and more. The reason for this is that the available datasets and prediction methods are on substantially different scales. Thus, while the CRISPRon is trained to minimize the MSE, the comparison to other models and on other datasets is necessarily done in terms of Spearman correlation. The MSE of CRISPRon and of the gradient boosting regressor tree trained on the same data are reported for all validation folds and on an internal independent test set in tables S7, S8, S9.

- (b)** We fully agree with the reviewer, and we discuss in Supplementary Note 1 how it is the addition of training data rather than the choice of the learning strategy that leads to improved predictions. Indeed, our gradient boosting regression tree presents performances close to those obtained by deep learning. Nevertheless, the comparison between CRISPRon and other available tools is done on the trained models, as these are the actual services offered to the users.

- (c)** The current predicted value is the indel frequency (%) of a gRNA. We believe that expressing efficiency in term of indel frequency is more informative and allows for a better prioritization of the gRNAs during the design phase, when multiple candidate gRNAs may be available for the same task. Also, predicting classes challenges the comparison to other available prediction models.

6. If there was substantial editing prior to Dox addition, then this editing could have occurred before lentivirus integrated into the genome. Thus, the data from this work does not necessarily represent genome-integrated constructs. The authors should clearly acknowledge this caveat. Ideally they would repeat all experiments that

showed substantial editing in the absence of Dox. But if not, this emphasizes again the need for independent validation of individual native genomic sites.

Re: We appreciate the reviewer for highlighting the potential of editing the lentiviral genome before integrating to the genomic sites. The cells indeed expressing substantial level of SpCas9 prior to Dox addition. We can clearly detect editing outcome at day 2 (prior to Dox addition). In this revision, we have provided data supporting that, despite this caveat, the gRNA activity data measured by our approach can well capture editing efficiency at the native genomic sites and are in well agreement with those measured by Kim et al. (2019) and Wang et al (2019) (see response to point 3). Besides, in another unpublished study based on the CRISPRTRAP-seq approach, we have again demonstrated that the high throughput approach by CRISPRTRAP-seq can faithfully capture CRISPR-Cas9 editing features.

Figure X1. This unpublished result is based on the CRISPRTRAP-seq approach to capture the PAM motif of CRISPR nucleases (SpCas9 presented here). The designed is based on 27-nt target sites. Instead of using the canonical PAM, we have introduced all combinations of 4mer NNNN following the protospacer. Our results faithfully capture the canonical NGG PAM motif of SpCas9, while in the meantime also recapitulate that relatively low editing efficiency from NAG and NGA PAMs.

7. Targeting native genes could cause biases if the native targets are essential or inhibit growth, as mutations in the native counterparts of these sites could impair cell growth. The authors should compare Cas9-nuclease and CBE stop-gain mutation efficiency in essential genes (Cancer DepMap) represented in their library to determine whether their results are confounded by native gene essentiality, and if so they should discuss the implications for their work.

Re: The suggestion of removing the essential genes (enrichment or depletion gRNAs) from the dataset is highly valuable. We have reanalyzed our data and carefully

removed genes and gRNAs that could inhibit growth or vice versa. With our stringent criterial, we are able to retain 10592 gRNA efficiency for analysis.

8. Because the authors do not design their library randomly for ABE and CBE testing, it is possible that they may not have equivalent representation of 3-mer sequences centered on editable bases, which would cause blind spots in their model. They should provide a supplementary figure or table that describes the representation of each 3-mer with an A or C in the center.

9. The authors should compare their ABE and CBE predictive models to the newly published BE-Hive model (Arbab, Shen et al Cell). While this did come out after I received this paper to review, it is too relevant for them not to compare algorithmic performance.

Re to point 8 and 9: Due to the published BE-Hive model, we have decided to focus on SpCas9 gRNA efficiency quantification, prediction and the development of the CRISPRon model. We will adapt the reviewer's suggestions and comments to the ABE and CBE, and develop another algorithm to more accurately and robustly capture base editors. We hope the reviewer agree with us that merging the base editor data into the current CRISPRon manuscript will significantly dilute the focus.

Minor points

1. Figure 2e shows a discrepancy of 1bp insertions among four nucleotides, and insertions are highly dependent on the N17 nucleotide (fig 2f). Methods mentions that features in the BRR include 604 one-hot encodings of nucleotide(s), leading me to think that among their gRNAs there may have been an overabundance of certain features, as in N17 nucleotide, given the findings shown in 2e and 2f. The authors should normalize Fig. 2e based on N17 base identity.

Re. We would like to point out that the reviewer probably had misunderstood the figure 2e and 2f. This is probably caused by the non-consistent color legend. Both Figure e and f consistently show the preference of inserting T, followed by A/C, and disfavor G. The numbers provided in 2f showed that there is no overabundance for the nucleotide at N17 position in our dataset. N17 (A) = 2575, N17 (C) = 2971, N17 (G) = 2873, N17 (T)=2349. The old panel e is no longer included in the revision.

Fig 1e-f from previous submission (just for reference).

2. The most significant features by SHAP analysis seem to be taking priority among these predictions without much regard for minor features that don't influence ordinality of the results. Thus, it would be worth asking whether their model changes

if a different accuracy metric is used other than r . As an example, the authors show that the motif percentages in N5-N7 are over represented in both high and low efficient guides (fig 4c). If this is going into the model as a feature, it might support the idea that a lot of these guides are taking on this feature weight that aligns with a high spearman correlation, but might not align with other measures of regression accuracy. Looking at the SHAP values, some of the most relevant features are related to GC count or other counts of G and C nucleotides. If CAG is showing up 35% of the time in positions N5-N7 in the extreme cases (less than 1% efficiency and over 20% efficiency), the adjacent nucleotides will give rise to a GC motif more often even if it's by chance (25% of the time). Since GC content is a big driver of the prediction according to SHAP, this could be largely influenced by the aforementioned GC issue that will arise in the extreme cases, where ~35% of the extreme cases are high in GC content, in training these examples will influence the weight of these parameters, and predictions on the gRNAs with this feature that are not in the high or low end of editing efficiency may suffer from the increased weight of this feature being bolstered. This would lead to what I explained previously in terms of things correlating in terms of spearman correlation, due to the same weight being added to their prediction, but may not be indicative of proximity to actual distance to the observed frequency. The authors should explain or account for this possible source of bias.

Re: We appreciate the reviewer's comments on assessing the motif percentages in N5-N7 for the base editor dataset. While this is no longer relevant for the current revised version of the manuscript, we take into account for developing the new base predictors. Regarding the over presentation of CAG is showing up 35% at the N5-N7 position for both high and low CBE editing gRNAs, this was due to the design of library. As mentioned in the methods, the gRNAs were retrieved from the iSTOP database, which were designed for converting CAG to TAG (stop codon) by CBE.

The new analysis of features importance in SpCas9 efficiency highlights that the binding energy feature ΔG_B is a key component in learning. This feature was firstly presented in the context of off-targets by the Gorodkin's group (Alkan et al., *Genome Biology* (2018)). Other important features are the melting temperatures and the nucleotide composition in the seed region, in line with previous reports (eg. Doench et al. *Nature Biotechnology* (2014) and Kim et al. *Nature Biotechnology* (2020)).

3. Observing the distribution of r values in fig S17 may indicate few points were going into this comparison, and the relatively uniform distribution among some of the figures doesn't support the claim that editing outcomes were correlated without providing levels of significance.

4. Comparisons to inDelphi predictive outcomes are not well described in terms of what exactly is being compared.

5. The analysis of Cas9 repair outcome distributions does not add any novelty to existing published work. As such, it should be considered whether this should be moved to the supplement.

Re (point 3-5): We would like to clarify the misunderstanding of Fig S17 by the reviewer. Data points included in the violin plot were already included in the figure as

“n” value. We have performed correlation analysis between the indel profile captured by CRISPRTRAP-seq and corresponding indel profile predicted by inDelphi. Thus, in the violin plot, there are actually over 10,000 data points (Pearson’s r values). We have rephrase the wording to more correctly reflecting the correlation, rather than the significance. We have added more elaboration to the comparison of indel profiles measured by CRISPRTRAP-seq to inDelphi (Supplementary note 4, Figure S7). We have also listed the indel figures as supplementary ones as suggested.

6. Typos

- a. Page 15 figure legend g misses the word THAT “between gRNAs THAT have”
- b. Page 15 figure d misses the “n” values marked as in figure g

Re: Typos have now been corrected.

Reviewer #2 (Remarks to the Author):

The manuscript by Xiang and colleagues describes a high-throughput method for assessing editing outcomes at 12,000 artificial sites using either Spy Cas, an adenine base editor, or a cytosine base editor. Using this approach, the authors were able to assess the efficiency at the edited sites as well as to deduce properties of each target site that could improve or hinder the likelihood of indel formation or base conversion. The data were also used to train a machine learning-based prediction tool that can be used for prediction of outcomes for additional sites.

Critique of the manuscript:

The claims made in lines 121-124 are not well-supported. If more data supporting these claims is not provided, then this section should be substantially re-phrased. The data presented in Figure S2 are so far removed from the claims made that it casts severe doubt on the entire manuscript.

The comparison of CBE outcomes at TRAP vs. endogenous sites (Fig. S2e) clearly demonstrates a substantial difference in editing efficiencies: 28 or 20% C>T conversion in the cassette data, but only 12 or 9% (respectively) in the genome. This suggests that the editing efficiencies at the two loci may substantially differ, even if the relative efficiencies are preserved (e.g. the first site is edited ~40% more efficiently than the second site). The ABE results suggest a good correlation, but there is inadequate data here to make any conclusions.

However, the correlation values cited in the paper – “(r2 = 0.96 – 0.99)” – are founded on a deeply flawed approach. Plotting TRAP site vs. genomic site correlations of all the positions in the relevant window (as is done in panels S2d+f), is an inexplicable choice that floods the analysis with irrelevant information. Plotting of all the bases that are not expected to change causes clusters of irrelevant data near “0%” and “100%” (the latter is for G positions in S2d, for T positions in S2f) which dominate the “best fit” line and will ensure a very high r2 value, regardless of the correlations we are actually interested in. Namely, the C>T sites in S2f, which are clear outliers, yet this critical finding is obscured by the pointless r2 value and is treated as non-

existent by the claims made in the paper (e.g. “The results validated that the CRISPR editing efficiency and outcomes from the surrogate sites were closely correlated”). In reference 27, the authors were inspecting indel formation, and their correlations (between endogenous and exogenous sites) are not particularly compelling: 0.65 to 0.82 with adjusting; $r = 0.52$ to 0.76 without adjusting. The portions of reference 29 that seem to be relevant (presumably Figure 1h and the related results) are not statistically powered (e.g. there are not replicates) and comparison of a single T7E1 assay gel to Sanger analysis of 100 clones is an “apples to oranges” situation with unclear informational value, especially regarding indel profiles. These two references do not support the claim that “a surrogate target site can faithfully recapitulate the endogenous editing efficiency and indel profile”. At best, the referenced work seems to demonstrate that surrogate sites can approximate endogenous editing events. The authors must address this glaring issue before this manuscript can be considered for publication. The best way to address this issue would be to perform deep sequencing of the same type (and analysis pipeline) used for the 12,000 sites that are the main focus of the paper. Using Sanger/ICE analysis for benchmarking of two locus pairs (ABE or CBE, TRAP site vs. genomic site) is not sufficient to provide compelling evidence that surrogate loci recapitulate editing outcomes. In addition to an appropriate method of detection/analysis, the authors are encouraged to perform multiple technical replicates per site as well as checking multiple locus pairs. The previous choice of CBE site (TYPM) was appropriate: it allowed assessment of both the overall editing efficiency as well as relative efficiency at sites within the base editor’s active window. More similar CBE loci should be examined, as well as multiple ABE loci with similar properties. The ABE locus (INHBC) allows comparison of overall efficiency but not relative efficiency at sites within the base editor’s active window.

Note that the above critiques apply to any time Fig. S2 is referenced (e.g. line 293).

Re (points above regarding Fig. S2 and the validation): We fully agree with the reviewer that the previous data presented by Sanger sequencing, ICE analysis and indel correction are inappropriate to draw the conclusion that the surrogate based CRISPRTRAP-seq method can well capture the indel frequency and outcomes as in the corresponding endogenous sites. Thus, we have now conducted more validations by deep sequencing, more loci included, and independent analyses and comparison to solidly address the solidarity of the CRISPRTRAP-seq approach for capturing the editing efficiency and outcomes.

In the revisions, we have performed deep sequencing of 17 endogenous loci and corresponding TRAP sites. Although the correlation between these 17 sites is 0.72 (Spearman, p value = 0.0016), the small data set should to some extent support that CRISPRTRAP-seq can to some extent capture the editing outcomes at the endogenous sites (Figure S3).

A

		TET2		indel	reads	freq.
		protospacer	PAM			
Endogenous locus	TTC	CAACAAGCTT	CAGTTCTACAGGGATAT	I1	397777	46.51
	TTC	CAACAAGCTT	CAGTTCTACAGGGATAT	D8	46887	5.48
	TTC	CAACAAGCTT	CAGTTCTACAGGGATAT	I1	37436	4.37
	TTC	CAACAAGCTT	CAGTTCTACAGGGATAT	D3	21473	2.51
	TTC	CAACAAGCTT	CAGTTCTACAGGGATAT	I1	17925	2.09
	TTC	CAACAAGCTT	CAGTTCTACAGGGATAT	D15	7504	0.87
	TTC	CAACAAGCTT	CAGTTCTACAGGGATAT	I1	5786	0.67
	TTC	CAACAAGCTT	CAGTTCTACAGGGATAT	I1	4632	0.54
	TTC	CAACAAGCTT	CAGTTCTACAGGGATAT	I1	4455	0.52
	TTC	CAACAAGCTT	CAGTTCTACAGGGATAT	D12	3221	0.37

		TET2		indel	reads	freq.
		protospacer	PAM			
TRAP locus	GCAGAAGTC	CAACAAGCTT	CAGTTCTACAGGGATAT	I1	58505	53.28
	GCAGAAGTC	CAACAAGCTT	CAGTTCTACAGGGATAT	D8	57877	5.52
	GCAGAAGTC	CAACAAGCTT	CAGTTCTACAGGGATAT	D3	32657	3.09
	GCAGAAGTC	CAACAAGCTT	CAGTTCTACAGGGATAT	I1	21629	2.06
	GCAGAAGTC	CAACAAGCTT	CAGTTCTACAGGGATAT	D15	12177	1.16
	GCAGAAGTC	CAACAAGCTT	CAGTTCTACAGGGATAT	D15	11370	1.08
	GCAGAAGTC	CAACAAGCTT	CAGTTCTACAGGGATAT	D8	5009	0.47
	GCAGAAGTC	CAACAAGCTT	CAGTTCTACAGGGATAT	D16	4966	0.47
	GCAGAAGTC	CAACAAGCTT	CAGTTCTACAGGGATAT	D5	4500	0.42
	GCAGAAGTC	CAACAAGCTT	CAGTTCTACAGGGATAT	I1	4373	0.41

		CAMK1G		indel	reads	freq.
		protospacer	PAM			
Endogenous locus	TCTGCTG	CGAAGTGC	TGGCCAGAACCCCTACAGCAAGCTGTGGATTGCTGGTCCATCGGCATCACCTAC	I1	239843	46.38
	TCTGCTG	CGAAGTGC	TGGCCAGAACCCCTACAGCAAGCTGTGGATTGCTGGTCCATCGGCATCACCTAC	D3	15269	2.95
	TCTGCTG	CGAAGTGC	TGGCCAGAACCCCTACAGCAAGCTGTGGATTGCTGGTCCATCGGCATCACCTAC	D2	9108	1.76
	TCTGCTG	CGAAGTGC	TGGCCAGAACCCCTACAGCAAGCTGTGGATTGCTGGTCCATCGGCATCACCTAC	I2	8831	1.70
	TCTGCTG	CGAAGTGC	TGGCCAGAACCCCTACAGCAAGCTGTGGATTGCTGGTCCATCGGCATCACCTAC	D11	6472	1.25
	TCTGCTG	CGAAGTGC	TGGCCAGAACCCCTACAGCAAGCTGTGGATTGCTGGTCCATCGGCATCACCTAC	D8	3647	0.70
	TCTGCTG	CGAAGTGC	TGGCCAGAACCCCTACAGCAAGCTGTGGATTGCTGGTCCATCGGCATCACCTAC	D20	3439	0.66
	TCTGCTG	CGAAGTGC	TGGCCAGAACCCCTACAGCAAGCTGTGGATTGCTGGTCCATCGGCATCACCTAC	D6	3388	0.65
	TCTGCTG	CGAAGTGC	TGGCCAGAACCCCTACAGCAAGCTGTGGATTGCTGGTCCATCGGCATCACCTAC	D42	3216	0.62
	TCTGCTG	CGAAGTGC	TGGCCAGAACCCCTACAGCAAGCTGTGGATTGCTGGTCCATCGGCATCACCTAC	D2	3192	0.61

		CAMK1G		indel	reads	freq.
		protospacer	PAM			
TRAP locus	CAGAAGTGC	TGGCCAGAACCCCTACAGCAAGCTGT	GGATTGCTGGTCCATCGGCATCACCTAC	I1	222904	44.52
	CAGAAGTGC	TGGCCAGAACCCCTACAGCAAGCTGT	GGATTGCTGGTCCATCGGCATCACCTAC	D16	18499	3.69
	CAGAAGTGC	TGGCCAGAACCCCTACAGCAAGCTGT	GGATTGCTGGTCCATCGGCATCACCTAC	D3	11883	2.38
	CAGAAGTGC	TGGCCAGAACCCCTACAGCAAGCTGT	GGATTGCTGGTCCATCGGCATCACCTAC	I2	10265	2.01
	CAGAAGTGC	TGGCCAGAACCCCTACAGCAAGCTGT	GGATTGCTGGTCCATCGGCATCACCTAC	D21	10279	2.01
	CAGAAGTGC	TGGCCAGAACCCCTACAGCAAGCTGT	GGATTGCTGGTCCATCGGCATCACCTAC	D11	8761	1.73
	CAGAAGTGC	TGGCCAGAACCCCTACAGCAAGCTGT	GGATTGCTGGTCCATCGGCATCACCTAC	D3	8640	1.73
	CAGAAGTGC	TGGCCAGAACCCCTACAGCAAGCTGT	GGATTGCTGGTCCATCGGCATCACCTAC	D8	7245	1.44
	CAGAAGTGC	TGGCCAGAACCCCTACAGCAAGCTGT	GGATTGCTGGTCCATCGGCATCACCTAC	D8	6652	1.33
	CAGAAGTGC	TGGCCAGAACCCCTACAGCAAGCTGT	GGATTGCTGGTCCATCGGCATCACCTAC	D2	5506	1.10

B

To further validate that the gRNA efficiency measured by CRISPRTRAP-seq is consistent with those measured by other studies, we compared the gRNA efficiency of overlapping gRNAs between CRISPRTRAP-seq and those from the Kim et al. (2019) study and those from the study of Wang et al. (2019). As shown in the figure below,

there is an equally good correlation for gRNA efficiency among the three independent studies (Figure 1g).

To even further demonstrate the validity of the CRISPRTRAP-seq method, we share some unpublished results which is not part of the current study but obtained based on the CRISPRTRAP-seq approach. This unpublished result is based on the CRISPRTRAP-seq approach to capture the PAM motif of CRISPR nucleases (SpCas9 presented here). The designed is based on 27-nt target sites. Instead of using the canonical PAM, we have introduced all combinations of 4mer NNNN following the protospacer. Our results faithfully capture the canonical NGG PAM motif of SpCas9, while in the meantime also recapitulate that relatively low editing efficiency from NGA and NAG PAMs (Fig X1, unpublished results from another ongoing study based on the CRISPRTRAP-seq approach). The method is developed for in-cell identification of PAM motifs.

Day 2 editing data seem to represent anomalous outcomes that may not represent typical genome editing outcomes and may instead reflect some sort of experimental artifact specific to the early time-point. Perhaps this is because the lentivirus itself is still present at Day 2. In particular, the Day 2 data in Fig. 1e (the pie chart) reveal a

pattern where “other” outcomes predominate: ~70% of outcomes are “other” on Day 2 but <20% on Day 8 and Day 10. Furthermore, the fact that Day 2 outcomes have a very poor correlation with inDelphi predictions (Fig. S17) suggests that something anomalous is happening. The authors must discuss these “other” outcomes in greater detail; what is their nature, and why might they appear so frequently in the Day 2 results? The Fig 1e caption suggests that these could be wild-type reads, which would be confusing if true, because there are no apparent wild-type (0% editing) reads in the Day 2 data of Fig. 1d. Please clarify what is responsible for the unexpected Day 2 results of Fig. 2e and Fig. S17. The current statements at lines 224-227 could also be updated; it seems unlikely that this shift can be attributed to “experimental conditions”. If there is any precedent for an observation like this (dramatic change in indel patterns), it should be discussed. I am not aware of any such precedent.

The above issues must be addressed in a revised manuscript before it can be seriously considered for publication.

Re: First of all, we would like to clear out one misunderstanding, probably we haven't described that clearly in the pie chart legend. The category of “other” are actually mainly WT reads together with a small fraction of indel reads (deletion larger than 30nt and insertion large than 10nt). That is why Day 2 data contain more “other” reads. Apology for this confusing, and we have addressed this in the revision.

The editing efficiency data at Day 2 is most likely a combination of both integrated and episomal lentiviral gDNA. The SpCas9 expressing cells that we have been using, although the expression cassette is based on a TRE promoter, substantial leakiness of SpCas9 expression is detected. Thus, the cell line was treated as a cell line with low-normal expression of SpCas9. High SpCas9 expression can be induced by Dox addition. Our data also clearly that most gRNA efficiency from the Dox addition groups are above 90%. This is also why for model establishment; we only used the gRNA efficiency from cells without Dox addition.

Regarding the indel dynamics, there is indeed articles from our colleagues demonstrating that depending the way of CRISPR delivery (plasmid, lentiviral, RNP etc), the indel profile can be varied. Besides, indel profiles introduced by lentiviral CRISPR vectors exhibit a change from early profile (e.g. 2 days after transduction) to a more stable indel profile (one week after transduction and stably integration of the vector to the cells). It is not surprisingly that the indel profile from Day2 correlated poorly with indel profiles predicted by inDelphi, as the in Delphi is built on the indel profiles from stable phase. That is also why we have better correction of indel profiles measured by CRISPRTRAP-seq and those predicted by inDelphi. This also provide a validation to our method.

It may be helpful to talk about the relationship between the data in Fig. 2f and 2g. Perhaps it would be reasonable to consider a model where most indel repairs are

driven by insertion of a base matching that at position N17, but in the case of a “G” at N17, deletion is favored. This may explain the weaker influence of “G” at N17 on insertions, as we see in Fig. 2f.

Re: Great thanks for highlighting this point and the suggestion to the model of insertion and deletion. Although in the revision, we have now focusing on delivering the best on-target efficiency predictor. As you can see from our benchmarking analysis, we indeed achieve this by using the merge data generated by our approach and those from the study of Kim et al. (2019). We still not quite understand the repair mechanism how the N17 nucleotide has such a strong influence on the insertion (as well as insertion type). But observation provide with this vast amount of data is clear, as we have harnessed this observed for many applications as well. For example, in one of our on-going study which are employing the CRISPR for knocking out HIF1A in HUVEC cells. As you can see from our TIDE analysis of **a pool of CRISPR edited cells**, one gRNA (HIF1A-gRNA1) predominantly create 1bp insertion “T”, which is exactly the same as the N17 nucleotide. We have included this predictive feature in our discussion (supplementary note 4). The figure below shows unpublished results from the Luo lab. Please note that data presented here is from a pool of CRISPR-edited cells, not from a single cell clone. Indel frequency was estimated by ICE analysis.

Consider using consistent colors in Fig. 2e+f for A/C/G/T. It is confusing that they are currently different. Supplemental figures would ideally also follow a single, consistent theme, so this would call for an update of the colors in Fig. 2e.

Re: Sorry for that. We have now corrected this in consistent colors.

Reconsider the name “TRAPseq” since an essentially identical term has already been established, e.g. “Translating Ribosome Affinity Purification (TRAP) Followed by RNA Sequencing Technology (TRAP-SEQ)”

Re: Thanks for pointing out this. We have now selected the name of CRISPRTRAP-seq.

Figure S5: Although many of the bars have error bars that are nearly invisible, analysis of the vector art demonstrates that at least one of these bars (library 2, 10 μ L)

has no error bars at all. Please make note of this, if it is fact an $n = 1$ condition.

Re: Figures have been thoroughly updated.

Supplementary Tables: Perhaps I missed this information, but it seems that there are no legends provided for the supplementary tables. If that is the case, it should be rectified. It is not clear what the values represent in Tables S3 & S4.

Re: S Tables have been thoroughly updated, with more information and explanation to values included in each supplementary table.

Throughout the text, it would be extremely valuable to replace all instances of “guide sequence” with “spacer sequence”. The “guide sequence” refers to the full RNA that Cas9 binds to, either in sgRNA or dgRNA format. The 5’ portion of the crRNA or sgRNA that determines what DNA sequence is being targeted is referred to as the “spacer” and it seems that this sequence is the focus of the secondary structure predictions in this work. Furthermore, the materials/methods section on this topic (section “vi” specifically) is currently incomprehensible regarding the secondary structure prediction that was performed. Please make this clear and provide adequate detail so the experimental procedures can be understood and potentially repeated.

Re: Thanks for pointing out the “spacer” and “guide” sequences. It is a mistake that we should not make. In the revision, we have clarified the term used and have tried to present them as accurate and precise as possible. Methods sections have been updated as well.

Regarding the interpretation of Figure 2a+b, it seems that the claims are not fully supported. There is no mention of statistically significant differences between different clusters/bins of data (e.g. 30-40% vs. 40-50% in 2a), so it is unclear how/why claims are being made regarding the “optimal” ranges.

Re: In the revision, we have added statistics to the different clusters/bins of data. We have also revised the conclusions. Nevertheless, in the revision, we have integrated all data and features and develop the CRISPRon predictor and gRNA designing tool for the CRISPR community.

“Importantly, according to our knowledge, this is the first time that both ABE and CBE efficiencies are measured at such a large scale in cells.”

I’m not sure if priority/novelty claims are allowed in this journal, but even if you are permitted to make such a claim, it may no longer be true due to the recent Arbab et al. publication.

Re: In this revision, we have substantially focusing on presenting the development of the CRISPRon prediction tool based on data gRNA efficiency data generated by CRISPRTRAP-seq and data from the study of Kim et al. (2019). We hope the reviewer agree with us that including the ABE/CBE data into the CRISPRon study might be not appropriate anymore. Besides, due to the publication from Arbab et al. while this manuscript was in review, simple presenting the ABE/CBE data will not provide sufficient novelty. Thus, we have focused on developing the deep learning CRISPRon gRNA efficiency predictor and the CRISPRon-based gRNA designing web tool. The ABE and CBE data will be more thoroughly analyzed, models

developed and comparison with Arbab et al. These will be presented in another study later.

Feedback on readability:

Fig. S38b Labels should read “stopped”
77 Should be “attractive tools”.

80-81 Base editors are described as able to create “a [...] substitution” within the editing window, but in many cases there are multiple edits within the window. Consider revision to “substitutions”.

82 Consider changing “Albeit” to “Even with”

83 Change to “urgent need for”, change “DBS” to “DSB”

57-87 The entire introduction has excessive focus on Cas9. Many statements that refer to editors in general are currently making reference to Cas9. Instead, consider referring to “genome editors”, “genome editing enzymes”, or “RNA-guided endonucleases (RGENs)” to be more inclusive of relevant technologies.

89-91 For many scientists (especially those working with animals), the term “in vitro” is used to include tissue culture experiments. To distinguish between cell-free (biochemical/enzymatic) assays and tissue culture assays, you should use other terminology.

93-4 Consider revising “we lack” to “there is a deficit of”. This will convey that we don’t have enough data. “Lack” suggests that we have no data, and currently there is at least one other report that contains large-scale ABE/CBE data.

115 Consider revising to “of at most 170 bp”

115 “170 bp” is not consistent with 37bp”. Use the former style (with a space) throughout, since all values should be separated from units by a space. Same issue on line 333 - be consistent.

131-2 Correct to “Figure not drawn to scale.”

199-200 Consider revising to “we were capable of analyzing the likely consequences of all indels on protein translation”. This change is important because sometimes the actual outcomes differ from predictions.

230 Should be “therefore”

234-235 Should this be “also see the CRISPR Atlas resource” or “also seen in the CRISPR Atlas resource”?

281 “GC content delta G energy” if this is meant to represent two distinct features, add a comma.

283-286 This sentence is hard to understand, especially regarding the “efficiency prediction...” clause , the “efficiency dataset” clause and their relationship to other clauses.

287 Consider revision to “has been deposited and is publicly available at GitHub”

290 Same suggestion as lines 93-4.

376 Should be “analyzed”

421 Should be “First, based on the optimized”

810 Should be “split” (“splitted” is not in use, typically)

906-907 “For the editing window is wider” - it seems that a word might be missing here. Check the entire sentence and make it more clear if possible.

Re (readability): We really appreciate the suggestions and corrections to the typos, readabilities, and terms used etc. In the revision, we have thoroughly addressed this. s

Reviewers' Comments:

Reviewer #1:

Remarks to the Author:

The claims of this manuscript have been pared down significantly, and appropriate validation data has been added as well as extensive comparison with existing approaches. This manuscript is suitable for publication.

Reviewer #2:

Remarks to the Author:

The revised manuscript is much improved and I believe this work is now generally suitable for publication. I continue to take issue with the use of any unique/distinct name for this approach, which is similar to work published previously. I believe that it is not necessary to coin a new term to describe this approach.

Reviewer #1 (Remarks to the Author):

The claims of this manuscript have been pared down significantly, and appropriate validation data has been added as well as extensive comparison with existing approaches. This manuscript is suitable for publication.

Re: We thank the reviewer again your valuable and constructive comments for the improvement of our manuscript and the study.

Reviewer #2 (Remarks to the Author):

The revised manuscript is much improved and I believe this work is now generally suitable for publication. I continue to take issue with the use of any unique/distinct name for this approach, which is similar to work published previously. I believe that it is not necessary to coin a new term to describe this approach.

Re: We thank the reviewer again your valuable and constructive comments for the improvement of our manuscript and the study. In this final revised version of the manuscript, we have removed the term for the surrogate approach in capturing CRISPR gRNA efficiency. Instead, we simply described the approach's working principle and the modifications introduced by our study.